# FnCas9-based CRISPR diagnostic for rapid and accurate detection of major SARS-CoV-2 variants on a paper strip

**Manoj Kumar[1,2†], Sneha Gulati[1,2†], Asgar H Ansari[1,2], Rhythm Phutela[1,2], Sundaram Acharya[1,2], Mohd Azhar[1,2], Jayaram Murthy[1,2], Poorti Kathpalia[1,2], Akshay Kanakan[1], Ranjeet Maurya[1,2], Janani Srinivasa Vasudevan[1], Aparna S[1], Rajesh Pandey[1,2], Souvik Maiti[1,2,3]\*, Debojyoti Chakraborty[1,2]\***

[1]CSIR-Institute of Genomics & Integrative Biology, Mathura, India; [2]Academy of Scientific & Innovative Research (AcSIR), Ghaziabad, India; [3]CSIR-National Chemical Laboratory, Pune, India

**\*For correspondence:**
souvik@igib.res.in (SM);
debojyoti.chakraborty@igib.in
(DC)

[†]These authors contributed
equally to this work

**Competing interest:** See
page 13

**Reviewing editor:** Yamuna
Krishnan, University of Chicago,
United States

**Abstract** The COVID-19 pandemic originating in the Wuhan province of China in late 2019 has impacted global health, causing increased mortality among elderly patients and individuals with comorbid conditions. During the passage of the virus through affected populations, it has undergone mutations, some of which have recently been linked with increased viral load and prognostic complexities. Several of these variants are point mutations that are difficult to diagnose using the gold standard quantitative real-time PCR (qRT-PCR) method and necessitates widespread sequencing which is expensive, has long turn-around times, and requires high viral load for calling mutations accurately. Here, we repurpose the high specificity of *Francisella novicida* Cas9 (FnCas9) to identify mismatches in the target for developing a lateral flow assay that can be successfully adapted for the simultaneous detection of SARS-CoV-2 infection as well as for detecting point mutations in the sequence of the virus obtained from patient samples. We report the detection of the S gene mutation N501Y (present across multiple variant lineages of SARS-CoV-2) within an hour using lateral flow paper strip chemistry. The results were corroborated using deep sequencing on multiple wild-type (n = 37) and mutant (n = 22) virus infected patient samples with a sensitivity of 87% and specificity of 97%. The design principle can be rapidly adapted for other mutations (as shown also for E484K and T716I) highlighting the advantages of quick optimization and roll-out of CRISPR diagnostics (CRISPRDx) for disease surveillance even beyond COVID-19. This study was funded by Council for Scientific and Industrial Research, India.

## Introduction

Testing and isolating SARS-CoV-2-positive cases has been one of the main strategies for controlling the rise and spread of the coronavirus pandemic that has affected about 157 million people and 3 million deaths all over the world (World Health Organization, WHO). Like all other viruses, the SARS-CoV-2 genome has undergone several mutations during its passage through the human host, and the positive or negative impact of most of them on virus transmission and immune escape are still under investigation (*Rabi et al., 2020*; *Pachetti et al., 2020*; *Korber et al., 2020*; *Zhou et al., 2021*).

In December 2020, health authorities in the United Kingdom had reported the emergence of a new SARS-CoV-2 variant (lineage B.1.1.7) that is phylogenetically distinct from other circulating strains in the region. With 23 mutations, this variant had rapidly replaced former SARS-CoV-2 lineages occurring in the region and has been established to have greater transmissibility through modeling and clinical correlation studies (*Chand, 2020*; *Horby et al., 2021*; *Davies et al., 2021*;

**eLife digest** SARS-CoV-2, the virus responsible for COVID-19, has a genome made of RNA (a nucleic acid similar to DNA) that can mutate, potentially making the disease more transmissible, and more lethal. Most countries have monitored the rise of mutated strains using a technique called next generation sequencing (NGS), which is time-consuming, expensive and requires skilled personnel. Sometimes the mutations to the virus are so small that they can only be detected using NGS. Finding cheaper, simpler and faster SARS-CoV-2 tests that can reliably detect mutated forms of the virus is crucial for public health authorities to monitor and manage the spread of the virus.

Lateral flow tests (the same technology used in many pregnancy tests) are typically cheap, fast and simple to use. Typically, lateral flow assay strips have a band of immobilised antibodies that bind to a specific protein (or antigen). If a sample contains antigen molecules, these will bind to the immobilised antibodies, causing a chemical reaction that changes the colour of the strip and giving a positive result. However, lateral flow tests that use antibodies cannot easily detect nucleic acids, such as DNA or RNA, let alone mutations in them.

To overcome this limitation, lateral flow assays can be used to detect a protein called Cas9, which, in turn, is able to bind to nucleic acids with specific sequences. Small changes in the target sequence change how well Cas9 binds to it, meaning that, in theory, this approach could be used to detect small mutations in the SARS-CoV-2 virus.

Kumar et al. made a lateral flow test that could detect a Cas9 protein that binds to a nucleic acid sequence found in a specific mutant strain of SARS-CoV-2. This Cas9 was highly sensitive to changes in its target sequence, so a small mutation in the target nucleic acid led to the protein binding less strongly, and the signal from the lateral flow test being lost. This meant that the lateral flow test designed by Kumar et al. could detect mutations in the SARS-CoV-2 virus at a fraction of the price of NGS approaches if used only for diagnosis. The lateral flow test was capable of detecting mutant viruses in patient samples too, generating a colour signal within an hour of a positive sample being run through the assay.

The test developed by Kumar et al. could offer public health authorities a quick and cheap method to monitor the spread of mutant SARS-CoV-2 strains; as well as a way to determine vaccine efficacy against new strains.

---

*Volz et al., 2021*). Subsequently, the South African and Brazilian authorities have also reported variants (B.1.351 and P.1, respectively) that have been associated with a higher viral load in preliminary studies (*Tegally et al., 2020*; *Faria et al., 2021*). Till May 2021, at least 13 emerging SARS-CoV-2 lineages with multiple mutations have been reported from sequencing efforts across the globe (*CDC, 2021*). Variants in lineages of the SARS-CoV-2 genome have been classified either as Variants of Concern (VOC) or Variants of Interest (VOI). Among these, the VOCs have shown an indication of higher severity, transmissibility, and impact on pandemic spread from ongoing studies. VOIs have limited evidence for the same currently but are being monitored for potential impact in the future. Till May 2021, at least five lineages (B.1.1.7, P.1, B.1.351, B.1.427, and B.1.429) have been classified as VOCs and are being investigated for their susceptibility to diagnosis, response to vaccines, and correlation with disease severity and transmission (*CDC, 2021*).

As air travel has resumed in most countries post lockdown, infected individuals harboring these mutations have been detected in countries far away from their origin, generating concerns about greater disease incidence unless these cases are recognized and isolated. That is particularly important as vaccine roll-out has been initiated in several countries, and the studies about vaccine efficacy against the new variants are only in their infancy.

Among the diagnostic methods that have been employed for identifying coronavirus infected cases, qRT-PCR has been the most widely adopted nucleic-acid-based test due to its ability to identify even low copy numbers of viral RNA in patient samples and has been considered as a gold standard in diagnosis (*Mackay, 2002*; *Carter et al., 2020*). However, probe-based qRT PCR assays rely on amplification of a target gene with real-time analysis of DNA copy numbers and are generally not suited for genotyping point mutations associated with novel CoV2 variants (*Afzal, 2020*; *Yin, 2020*; *Vandenberg et al., 2021*). Although strategies such as Amplification Refractory Mutation System-

quantitative PCR (ARMS-qPCR) have been described in the literature for genotyping mutations in nucleic acids, these require substantial design and validation for reproducible readouts and hence are not routinely used for clinical diagnosis (*Little, 2001*). Similarly, antigen-based SARS-CoV-2 assays which have lower sensitivity than qPCR have not yet been reported to specifically identify mutant variants. In the absence of any true diagnostic test for the variants, most countries have resorted to next-generation sequencing of patient samples both for understanding the evolution of mutations in the virus as well as detecting the variants in suspected individuals (*Vandenberg et al., 2021*). Using sequencing for routine diagnosis greatly increases both the time and cost of identifying positive individuals (*Jayamohan et al., 2021*).

We and several others have documented the applicability of CRISPR diagnostics (CRISPRDx) for identifying CoV2 signatures in patient samples (*Broughton et al., 2020*; *Patchsung et al., 2020*; *Ding et al., 2020*; *Joung et al., 2020*; *Fozouni et al., 2021*; *Guo et al., 2020*; *Lucia et al., 2020*; *Wang et al., 2021*; *Azhar et al., 2021*). Using the Cas9 from *Francisella novicida* (FnCas9), we have recently reported the development of the FELUDA (FnCas9 Editor Linked Uniform Detection Assay) platform for SARS-CoV-2 diagnosis with similar accuracy as the gold standard qPCR test (*Azhar et al., 2021*). The assay which is performed on a Lateral Flow strip generates a visual readout within an hour that is quantifiable using a smartphone-based application. Since FnCas9 has a very high intrinsic specificity to point mismatches in the target (*Acharya et al., 2019*), we speculated that the enzyme can also be used for detecting SARS-CoV-2 variants on a paper strip with high accuracy. In this report, we introduce RAY (Rapid variant AssaY), a paper-strip based platform to identify mutational signatures of the coronavirus variants in a sample eliminating the need for sequencing-based diagnostics. RAY can successfully detect both SARS-CoV-2 infection as well as the presence of the common N501Y mutation present across the majority of VOCs described so far and distinguish it from the parent CoV2 lineage. Thus, it can be employed as a primary surveillance method for isolating cases belonging to these groups and can enable the initiation of appropriate management protocols.

## Materials and methods

### Study design
The study was designed to evaluate the efficacy of RAY on left-over patient samples. The intent of the study was to develop a robust CRISPR diagnostic that can perform with high accuracy for variant detection at significantly low cost and time taken for detection as compared to sequencing. For the N501Y detection using RAY, SARS-CoV-2 and control RNA samples were received from the diagnostic laboratory at CSIR-Institute of Genomics and Integrative Biology.

### Oligos
A list of all oligos (Merck) used in the study can be found in *Supplementary file 4*.

### Protein purification
Plasmids containing FnCas9-WT and dFnCas9 (catalytically-inactive, dead) (*Acharya et al., 2019*) sequences were transformed and expressed in *Escherichia coli* Rosetta 2 (DE3) (Novagen). Rosetta 2 (DE3) cells were cultured at 37°C in LB medium (with 50 mg/ml Kanamycin) and induced when OD600 reached 0.6, using 0.5 mM isopropyl β-D-thiogalactopyranoside (IPTG). After an overnight culture at 18°C, *E. coli* cells were harvested by centrifugation and resuspended in a lysis buffer (20 mM HEPES, pH 7.5, 500 mM NaCl, 5%) glycerol supplemented with 1X PIC (Roche) containing 100 μg/ml lysozyme. Subsequent cell lysis by sonication, the lysate was put through to Ni-NTA beads (Roche). The eluted protein was further purified by size-exclusion chromatography on HiLoad Superdex 200 16/60 column (GE Healthcare) in a buffer solution with 20 mM HEPES pH 7.5, 150 mM KCl, 10% glycerol, 1 mM DTT. The purified proteins were quantified by Pierce BCA protein assay kit (Thermo Fisher Scientific) and stored at −80°C until further use.

### RAY crRNA and primer design
All 48 VOCs/VOIs within 12 emerging SARS-CoV-2 lineages till May 2021 were taken from the Centers for Disease Control and Prevention report (*CDC, 2021*; *Rambaut et al., 2020*; *Hadfield et al.,*

2018). Further, by having these mutations at 2nd/6th/16th/19th bp upstream to PAM (NGG) in the SARS-CoV2- genome, a total of 19 among 48 variants could be targeted with the RAY strategy. Next, within the variant nucleotide containing crRNA sequence, a second synthetic mismatch nucleotide at the corresponding 6th/2nd/19th/16th position was added. Primers required for IVC assays flanking these crRNAs were designed using Primer3plus python library (*Untergasser et al., 2012*). Finally, the crRNAs/primers were checked for off-targets on a representative bacterial genome database (NCBI), virus genome database (NCBI) (*Brister et al., 2015*), and human genome/transcriptome (GENCODE GRCh38) (*Frankish et al., 2019*).

For the benefit of end users with quick design and implementation of the RAY for any targetable SNV/VOC/VOI, we now have also optimized our previously developed web tool JATAYU (Junction for Analysis and Target Design for Your FELUDA assay) (*Azhar et al., 2021*), which functions by incorporating mismatches in the sequence provided by the user. The server generates sgRNA and flanking primer sequences ready for synthesis (http://jatayu.igib.res.in).

## *In vitro* cleavage assay for 2 and 6 or 16 and 19 mismatched positions

Purified PCR 228 bp amplicon from *HBB* gene was used as substrate in *in vitro* cleavage assays, as optimized in our previous studies (*Acharya et al., 2019*). Substrate amplicons were treated with reconstituted dFnCas9 RNP complex (100 nM) in a reaction buffer (20 mM HEPES, pH7.5, 150 mM KCl, 1 mM DTT, 10% glycerol, 10 mM $MgCl_2$) at 37°C for 10 min, the cleaved products were visualized on a 2% agarose gel and densitometric quantification of images were done (*Figure 1A*).

> *HBB* WT sgRNA: GTAACGGCAGACTTCTCCTC
> *HBB* 2 and 6 MM sgRNA: GTAACGGCAGACTTATCCAC
> *HBB* 16 and 19 MM sgRNA:GGAAAGGCAGACTTCTCCTC

## RAY detection assays

Sequences containing WT and Mutant were reverse transcribed, and amplified using primers with/without 5' biotinylation from SARS-CoV-2 Viral RNA-enriched samples.

## Detection *via In vitro* Cleavage (IVC)

Purified PCR amplicons containing WT and Mutant sequences were used as substrates in *in vitro* cleavage assays, as optimized in our previous studies (*Azhar et al., 2021*). Substrate amplicons were treated with reconstituted RNP complex (100 nM) in a reaction buffer (20 mM HEPES, pH7.5, 150 mM KCl, 1 mM DTT, 10% glycerol, 10 mM $MgCl_2$) at 37°C for 10 min, and cleaved products were visualized on a 2% agarose gel.

## RAY *via* lateral flow assay (RT-PCR)

A region from the SARS-CoV-2 *S* gene containing N501Y/T716I/E484K mutations was reverse transcribed and amplified using a single end 5' biotin-labeled primer. *In vitro* transcription of sgRNAs/crRNAs was done using MegaScript T7 Transcription kit (ThermoFisher Scientific) following manufacturer's protocol and purified by NucAway spin column (ThermoFisher Scientific). Chimeric gRNA (crRNA:TracrRNA) was prepared by equally (crRNA:TracrRNA molar ratio,1:1) combining respective crRNAs and synthetic 3'-FAM-labeled TracrRNA in an annealing buffer (100 mM NaCl, 50 mM Tris-HCl pH8 and 1 mM $MgCl_2$) by heating at 95°C for 2–5 min and then allowed to cool at room temperature for 15–20 min. RNP complex was prepared by equally mixing (Protein:sgRNA molar ratio,1:1) Chimeric gRNA and dead FnCas9 in a buffer (20 mM HEPES, pH7.5, 150 mM KCl, 1 mM DTT, 10% glycerol, 10 mM $MgCl_2$) and rested for 10 min at RT. Target biotinylated amplicons were then treated with the RNP complexes for 10 min at 37°C. Finally, 80 µl of Dipstick buffer was added to the mix along with Milenia HybriDetect one lateral flow strip. The strips were allowed to stand in the solution for 2–5 min at room temperature and the result was observed. The strip images can be processed using the TOPSE application that generates background corrected values from the smart phone acquired images of the strip. This application has been previously trained on a large number of CoV-2 samples (*Azhar et al., 2021*). *In vitro* synthesized crRNAs used for double amplicon RAY were a gift from TATA Medical and Diagnostics. Detailed description of the protocol can be found in Appendix 1.

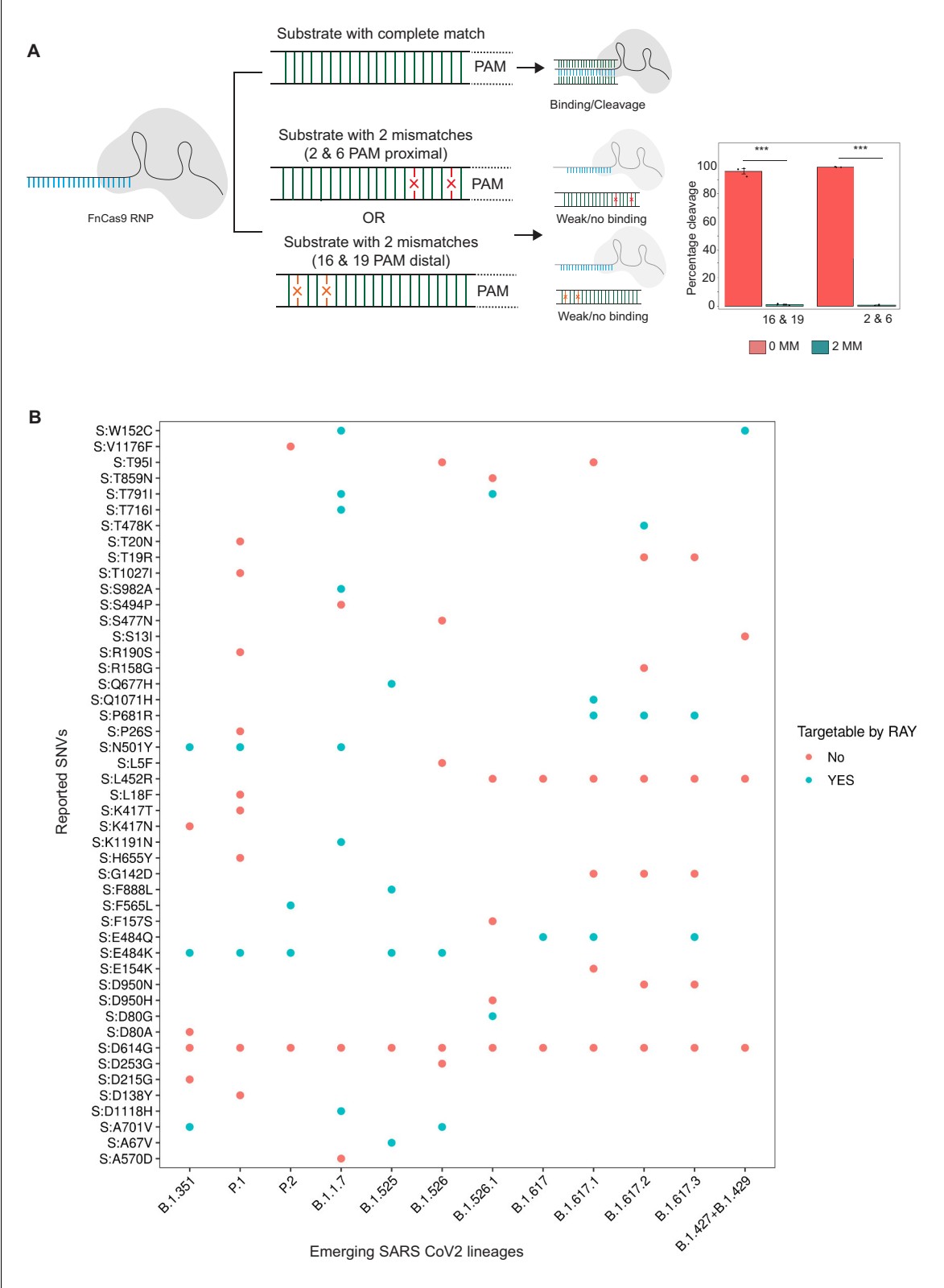

**Figure 1.** Schematic for RAY. (**A**) FnCas9 is unable to bind or cleave targets having two mismatches at the PAM proximal 2nd and 6th or PAM distal 16th and 19th positions as shown (left panel). The quantification of cleavage with a substrate with mismatches at indicated positions is shown (right panel, n = 3 independent experiments, errors s.e.m, student's paired T-test p values \*\*\*<0.001 are shown). (**B**) Dot plot showing the major SNVs (y-axis) present in the emerging SARS-CoV-2 lineages (x-axis). The status of each SNV as being targetable by RAY is indicated as dots.

## qRT-PCR SARS-CoV-2 detection

qRT-PCR was performed using STANDARD M nCoV Real-Time Detection kit (SD Biosensor) as per manufacturer's protocol. Briefly, per reaction 3 µl of RTase mix and 0.25 µl of Internal Control A was added to 7 µl of the reaction solution. Five µl of each of the negative control, positive control, and patient sample nucleic acid extract was added to the PCR mixture dispensed in each reaction tube. The cycling conditions on the instrument were as follows: Reverse transcription 50℃ for 15 min, Initial denaturation 95℃ for 1 min, 5 Pre-amplification cycles of 95℃ for 5 s; 60℃ for 40 s followed by 40 amplification cycles of 95℃ for 5 s; 60℃ for 40 s. Signal was captured in the FAM channel for the qualitative detection of the new coronavirus (SARS-CoV-2) *ORF1ab* (RdRp) gene, JOE (VIC or HEX) channel for *E* gene, and CY5 channel for internal reference.

## Sequencing of patient samples

Individuals who were found to be RT-PCR positive for SARS-CoV-2, were sequenced to determine whether it was a UK variant (20I/501Y.V1) or non-UK variant. With a low number of samples arriving each day in the lab and with quick turnaround time for reporting, Nanopore sequencing was undertaken for most of the samples.

## SARS-CoV-2 whole genome sequencing using nanopore platform

In brief, 100 ng total RNA was used for double-stranded cDNA synthesis by using Superscript IV (ThermoFisher Scientific, Cat.No. 18091050) for first strand cDNA synthesis followed by RNase H digestion of ssRNA and second strand synthesis by DNA polymerase-I large (Klenow) fragment (New England Biolabs, Cat. No. M0210S). Double stranded cDNA thus obtained was purified using AMPure XP beads (Beckman Coulter, Cat. No. A63881). The SARS-CoV-2 genome was then amplified from 100 ng of the purified cDNA following the ARTIC V3 primer protocol. Sequencing library preparation consisting of End Repair/dA tailing, Native Barcode Ligation, and Adapter Ligation was performed with 200 ng of the multiplexed PCR amplicons according to Oxford Nanopore Technology (ONT) library preparation protocol-PCR tiling of COVID-19 virus (Version: PTC_9096_v109re-vE_06Feb2020). Sequencing in sets of 24 barcoded samples was performed on MinION Mk1B platform by ONT.

## Nanopore sequencing analysis

The ARTIC end-to-end pipeline was used for the analysis of MinION raw fast5 files up to the variant calling. Raw fast5 files of samples were basecalled and demultiplexed using Guppy basecaller that uses the base calling algorithms of Oxford Nanopore Technologies (https://community.nanopore-tech.com) with phred quality cut-off score >7 on GPU-linux accelerated computing machine. Reads having phred quality scores less than seven were discarded to filter the low-quality reads. The resulting demultiplexed fastq were normalized by read length of 300–500 (approximate size of amplicons) for further downstream analysis and aligned to the SARS-CoV-2 reference (MN908947.3) using the aligner Minimap2 v2.17 (*Li, 2018*). Nanopolish (*Loman et al., 2015*) was used to index raw fast5 files for variant calling from the minimap output files. To create consensus fasta, bcftools v1.8 was used over normalized minimap2 output.

# Results

## RAY is able to target at least one SNV in every emerging lineage of SARS-CoV-2

In an earlier study, we had successfully established that FnCas9 is unable to bind or cleave targets having two mismatches at the 2nd and 6th position (PAM proximal) of the sgRNA with respect to the target (*Azhar et al., 2021*; *Figure 1A*). Through *in vitro* cleavage studies of a double-stranded DNA substrate, we identified that the same outcome is also observed when mismatches are present at the 16th and 19th position (PAM distal) (*Figure 1A*, Materials and methods). This implies that if a mismatch exists in any of these position combinations, placing an additional synthetic mutation in the sgRNA at the other positions makes FnCas9 unable to bind or cleave the target (*Figure 1A*). Thus, mutations at any of these positions relative to a NGG PAM site in the SARS CoV-2 variants could be potentially distinguished from the parent strain.

To develop RAY for identifying SARS-CoV-2 variants, we first analyzed the mutations arising in all SARS-CoV-2 lineages reported so far (May 2021) and looked for the possibility of FnCas9-mediated targeting based on the presence of an NGG PAM site in the vicinity. We found that out of the 48 unique SNVs reported across all the VOCs and VOIs reported so far (*CDC, 2021*), 19 SNVs (12 VOC-associated SNVs and 7 VOI-associated SNVs) can be targeted by RAY (*Supplementary file 1*). Interestingly, each VOC/VOI reported so far consists of at least one of the 19 SNVs and can therefore be targeted by RAY design (*Figure 1A*). Among these SNVs, N501Y is present across 3/5 of the VOCs (B.1.1.7, P.1, and B.1.351). This mutation affects the receptor-binding domain of the virus and is the subject of numerous studies analyzing the efficacy of vaccines against the mutated form of the virus (*Cheng et al., 2021*; *Dejnirattisai et al., 2021*; *Ku et al., 2021*; *Luan et al., 2021*; *Naveca et al., 2021*; *Tian et al., 2021*; *Xie et al., 2021*; *Collier et al., 2021*). Owing to its prevalence as a shared marker for the majority of mutated lineages, we took this SNV further for developing a diagnostic assay.

We designed primer pairs surrounding the N501Y mutation after analyzing the mutational spectrum in SARS-CoV-2 strains obtained from the publicly available sequencing database, GISAID (*Shu and McCauley, 2017*). Additionally, we ensured that no regions from human or non-human genome as well as transcriptome shared significant homology to the sites to prevent any non-specific amplification during the reverse transcription PCR (RT-PCR) reaction (Materials and methods). In our earlier study, we have validated two FnCas9 sgRNAs in the *N* and *S* gene of SARS-CoV-2 which could detect positive cases with high sensitivity and specificity (*Azhar et al., 2021*). We reasoned that including the *S*-gene sgRNA which lies in the vicinity of the N501Y variant would serve as an internal positive control both for the presence of the SARS-CoV-2 virus in the sample as well as quality control for the amplicons generated in the RT PCR step of the assay.

## RAY can successfully discriminate N501Y and WT nCoV2 substrates

We tested the N501Y sgRNA containing the variant mismatch at PAM proximal 2nd position to selectively bind and cleave the mutant substrate, while not affecting the WT substrate due to mismatch at PAM proximal 2nd and 6th positions. We found that catalytically active FnCas9 was able to successfully cleave the dsDNA substrate containing the N501Y mutation while leaving the WT sequence intact (*Figure 2A*). Importantly, this leads to a distinct pattern on agarose gel that can distinguish the two variants. We established that this approach for variant identification can be performed using a stand-alone or portable electrophoresis apparatus leading to a turnaround time of about 1.5 hr from RNA to read-out (*Figure 2A*).

The electrophoresis based identification of the N501Y(A23063T) mutation can be adapted for COVID-19 detection under laboratory conditions. However, dye-based systems have inherent low sensitivity and cannot be used for routine diagnosis. We reasoned that for RAY to detect and diagnose variants, we need to adapt the detection to a more sensitive readout. In an earlier study, we had reported that using a lateral flow assay, a catalytically inactive (dead) FnCas9 generated by mutating the RuvC and HNH domains can be used to accurately identify the presence of SARS-CoV-2 RNA in patient samples (*Azhar et al., 2021*). To enable such a readout, the RNA sample first undergoes reverse transcription PCR using biotin-labeled primers, generating labeled amplicons at the end of the PCR reaction. The products from the reaction are then incubated with dFnCas9: sgRNA ribonucleoprotein complex where the sgRNA component is labeled with Fluorescein amidite (FAM) at the 3′ end. Upon identifying its target, the RNP forms a tight complex with the dsDNA substrate. This complex is then applied on a paper strip that has gold nanoparticles tagged with antibodies recognizing the FAM label. In an aqueous environment, the nanoparticle:RNP:substrate complex moves up the strip by lateral flow and upon reaching a streptavidin line on the strip these get deposited if the RNP is attached to biotin-labeled substrates (test band). Residual or unbound AuNP complexes get deposited at a control line where secondary antibodies recognizing the anti-FAM antibody attached to the AuNPs are immobilized on the strip (*Figure 2B,C*).

We reasoned that in order to enable RAY to distinguish two samples different by a single mismatch on a paper strip, the intensity of the one mismatched sgRNA (as the other mismatch corresponds to N501Y SNV) should be several folds higher than wild-type samples (having two mismatches with the sgRNA, one at the SNV position and another synthetic mismatch). To generate a visually distinctive signal between WT and mutant sample, we performed a single-step reverse transcription PCR to generate a biotin-labeled amplified product that can be detected by a single

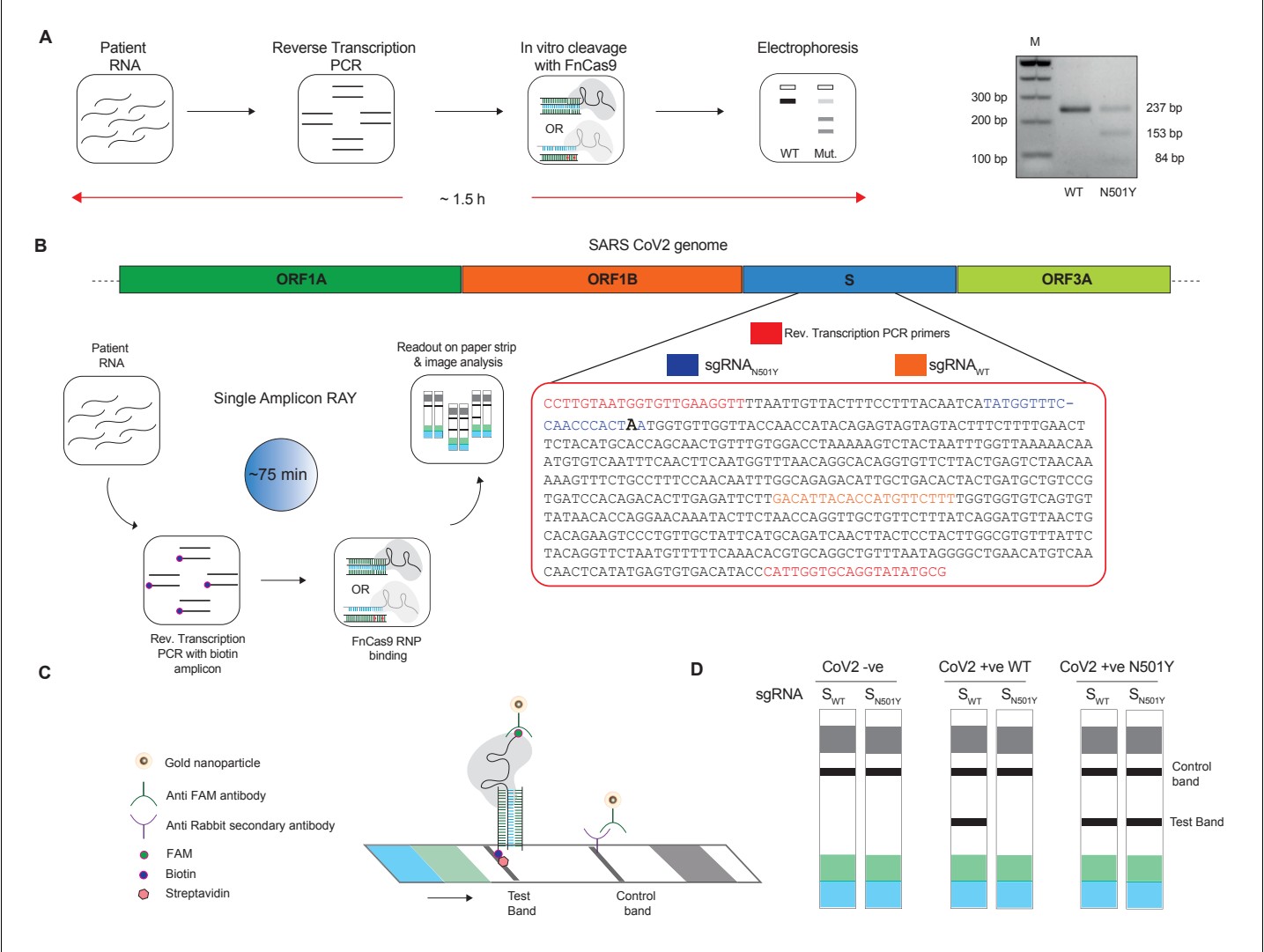

**Figure 2.** Adaptation of RAY for identification of N501Y mutation. (**A**) Schematic showing the application of RAY for distinguishing variants WT and N501Y using electrophoresis, key steps in the process are shown. The uncleaved band represents the amplicon from either the WT or N501Y sample, cleavage occurs only in the N501Y sample. (**B**) Schematic for detecting the N501Y through a single amplicon RAY on a lateral flow assay is shown. Left panel depicts key steps in the assay. The amplicon sequence and position of primers and sgRNAs are indicated on the right. The N501Y mutation position is shown in raised font. (**C**) Outcome of the association of FAM-labeled FnCas9 RNP bound to biotin-labeled substrate on the paper strip is shown. Arrow indicates the direction of flow. (**D**) Different outcomes of RAY on a paper strip based on the starting material. Distinct bands on the streptavidin line (test line) characterize CoV-2 negative, CoV-2 wild type and CoV2 N501Y variants.

sgRNA (called $S_{WT}$) if the sample is WT and by both sgRNAs ($S_{WT}$ and $S_{N501Y}$) if the sample contains the N501Y variant (*Figure 2B,C*). The presence of the $S_{WT}$ band also serves as a validation for the sample to be SARS-CoV-2 positive.

## Single Amplicon RAY can discriminate N501Y and WT nCoV-2 substrates from patient samples with high viral load

We first generated a single amplicon labeled with biotin at one end and performed RAY with both sgRNAs ($S_{WT}$ and $S_{N501Y}$) (*Figure 3A*). We reasoned that labeling biotin in the reverse primer would reduce possible background signal due to non-specific interactions of the unused biotin primers with the streptavidin line on the strip and increase the signal resolution between the mutant and WT samples. Among the four amplicon sizes that we investigated, an amplicon of 580 bp gave a clear discernible signal distinguishing the N501Y mutation over the WT sample (*Figure 3A,*

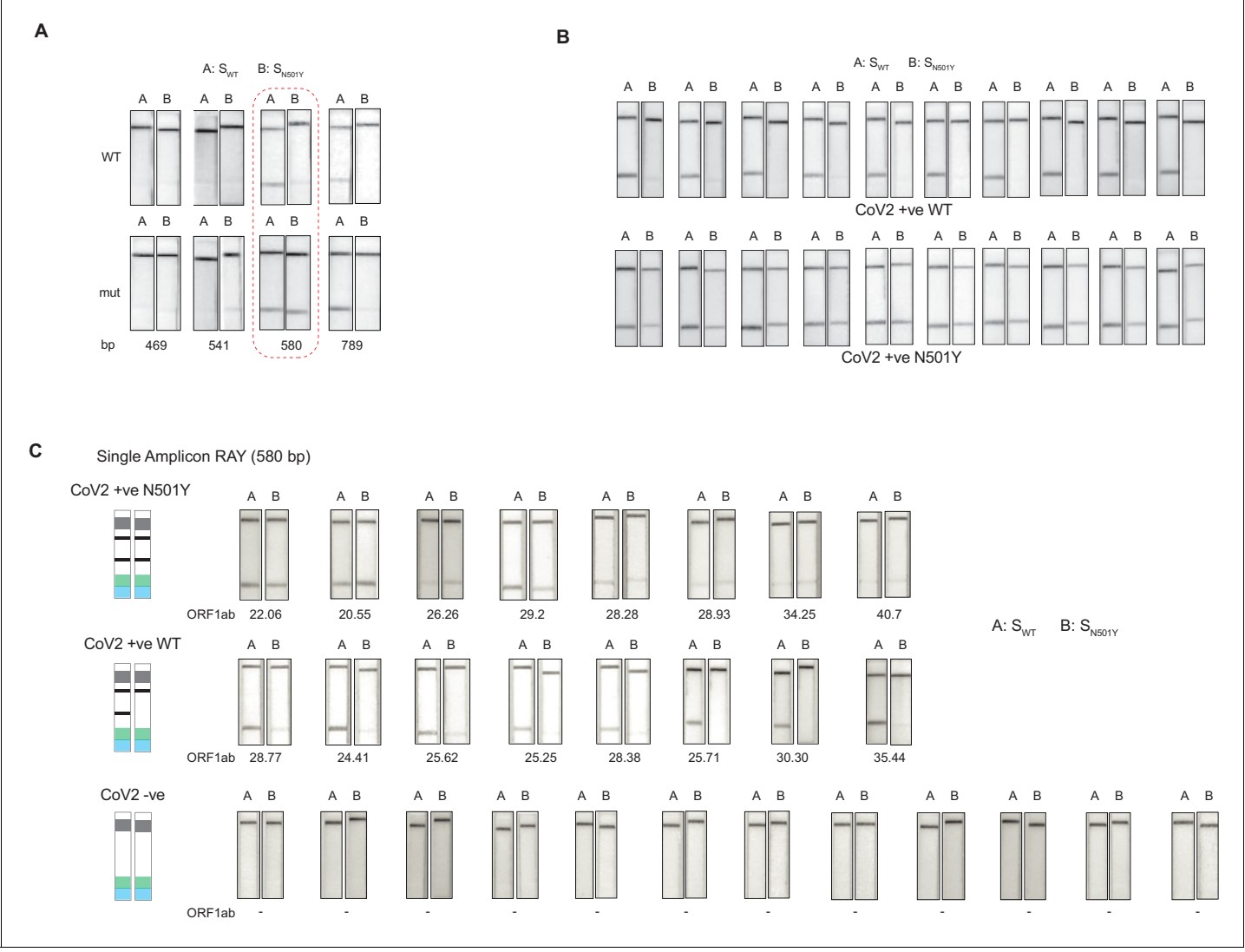

**Figure 3.** Validation of single amplicon RAY on patient samples. (**A**) RAY optimization with different size of S gene PCR amplicons. Optimal amplicon length denoted by red dotted box. (**B**) Reproducibility in output on multiple runs of RAY on the same samples (WT or N501Y) showing high concordance between assays (n = 10 RAY replicates from same sample). (**C**) RAY outcomes on three groups of patient samples as indicated. The *ORF1ab* Ct values for every sample is indicated below.

Materials and methods). To validate the reproducibility of this method, we took mutant and WT substrates, and performed RAY 10 times, and were able to successfully distinguish them on every occasion (*Figure 3B*).

Next, we tested RAY on RNA extracted from samples of eight qRT-PCR-positive SARS-CoV-2-infected individuals who harbored the N501Y mutation (along with other mutations). The samples were sequenced in parallel to detect the presence of N501Y mutation. RAY was able to correctly identify the variant signature in all eight samples that harbored the mutation (*Figure 3C*). However, we observed variability of signal intensities across the different samples. In particular, samples with low viral load (qRT-PCR Ct >25) showed a comparatively faint band (*Figure 3C*). This suggested that the RAY protocol required further optimization to increase the signal intensity, especially for low viral titres. Importantly, RAY correctly classified all eight WT samples where neither the N501Y mutation nor any other lineage variants were present as seen by either an absence of a distinct band or an extremely faint band in the test line (*Figure 3C*). In addition, RAY identified all confirmed COVID-19-negative samples (12) tested simultaneously (*Figure 3C*). Taken together, the single amplicon RAY

assay showed good specificity in identifying WT and SARS-CoV-2-negative samples but required improvements to increase the sensitivity for samples with low viral load.

## Double Amplicon RAY can successfully achieves high specificity and sensitivity with patient samples

To improve the performance of the assay for patient samples across a wide range of viral titres, we modified several aspects of the RAY assay. Firstly, we used TOPSE (True Outcome Predicted via Strip Evaluation), a smartphone-based application to generate a band intensity score from a lateral flow strip to eliminate bias that can result from visual estimation (Appendix 1). Secondly, we labeled both forward and reverse primers in the assay with 5' Biotin to increase the signal intensity and reduced the amplicon length of the substrate to get a consistent amplification at the end of each PCR run (*Figure 4A,B*).

We observed that upon double biotinylation the test band intensity for the 580 bp amplicon was 1.5 times higher than single biotin-labeled amplicon (*Figure 4A*). To increase the band intensity further, we reduced the length of the PCR amplicon containing the N501Y mutation and found that within the same sample, amplicons with shorter lengths had higher band intensities (*Figure 4B*). In particular, with an amplicon size of 149 bp, a clearly discernible background-corrected signal intensity could be achieved in the mutant sample over the WT sample (*Figure 4B*). Since the 149 bp amplicon did not contain the $S_{WT}$ sgRNA-binding site, we modified the assay to perform two simultaneous PCR reactions from the same sample, one generating the N501Y specific amplicon (149 bp), and another corresponding to the $S_{WT}$ sgRNA from the original FELUDA assay (287 bp) (*Figure 4C*). The double amplicon RAY assay results in shorter products and lower overall time for completion, starting from RNA samples while compared to a single amplicon RAY (reduced to 55 min from 75 min).

Next, we investigated the sensitivity of the assay on serial dilutions of a WT and a N501Y mutant patient sample with moderately high viral load (Ct <25). We observed that RAY was able to give a clearly discernible $S_{N501Y}$ sgRNA signal that was at least 5.5 times higher than that of the WT sample up to a Ct value of 34. Upon further dilution, no signal was observed both in RAY or qRT PCR for these samples (*Figure 4E*). Taken together, RAY is sufficiently sensitive in detecting the N501Y VOC in a sample with only a few copies of the virus and is comparable to other sequencing platforms (*Klempt et al., 2020*).

We then proceeded to test RAY on a set of patient samples containing either the WT (n = 37) or the N501Y variants (n = 22) identified by sequencing. For each of these samples, a qRT-PCR and the $S_{WT}$ RAY were first performed to confirm SARS-CoV-2 positivity (Appendix 1, *Supplementary file 1*). With a single run of the assay, $S_{N501Y}$ was able to detect 36/37 WT samples and 19/22 N501Y samples corresponding to a sensitivity of 86% and a specificity of 97% across all ranges of Ct values (*Figure 4F*, *Supplementary file 2*). Thus, RAY can identify the N501Y SNV with high accuracy in patient samples underscoring its impact as a rapid screening methodology for SARS-CoV-2 VOCs.

Notably, we were able to successfully validate RAY for two more mutations E484K and T716I in patient samples using corresponding sgRNAs suggesting that the assay can be optimized and adapted for other VOCs in addition to N501Y (*Figure 4G*). Among these, the E484K mutation has been associated with high transmissibility and the possibility of reinfection (*Weisblum et al., 2020*; *Xie et al., 2021*; *Collier et al., 2021*; *Wibmer et al., 2021*; *Annavajhala et al., 2021*; *Resende et al., 2021*; *Nonaka et al., 2021*; *Naveca et al., 2021*). Importantly, amplicons containing the E484K mutation do not show cross-reactivity with the N501Y sgRNA, highlighting the specificity of the assay between variants (*Figure 4H*). Finally, as seen in our earlier study, synthetic sgRNA containing a phosphorothioate backbone modification leads to higher stability and greater band intensity when compared to *in vitro* synthesized sgRNAs (*Figure 4I*).

## Discussion

Currently, deep-sequencing of patient samples is being routinely done by health authorities in various countries to track and isolate cases containing the SARS-CoV-2 variants (*Meredith et al., 2020*; *Oude Munnink et al., 2020*; *McNamara et al., 2020*; *Rockett et al., 2020*; *Bhoyar et al., 2021*). The accuracy of variant calling in any sequencing platform is dependent on the very high quality and coverage of reads to the reference genome. Since the process from sample to sequence consists of

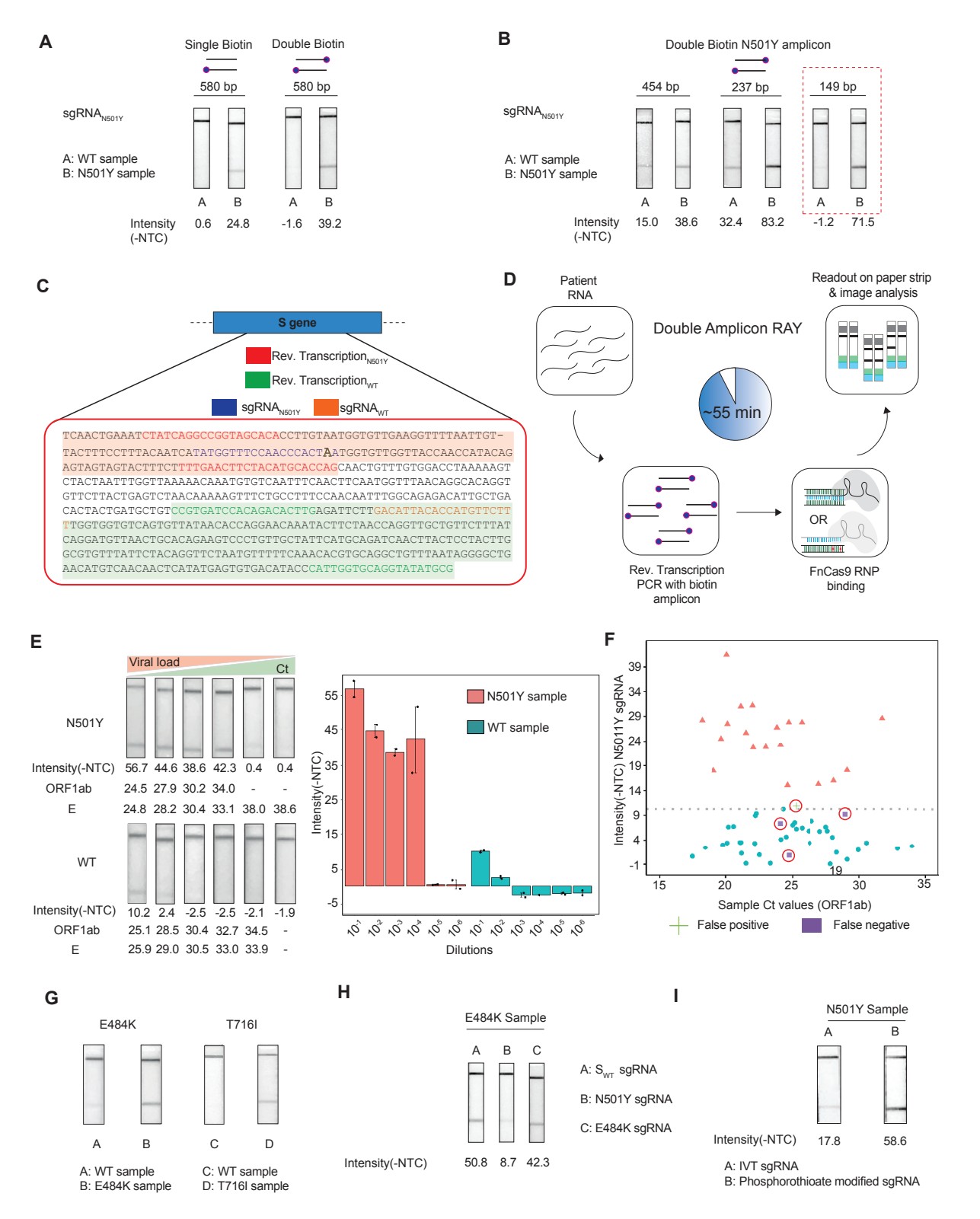

**Figure 4.** Double amplicon RAY. (**A**) Effect of single/double biotin labeling on primers for the 580 bp amplicon. (**B**) Effect of reduction of amplicon length on RAY outcomes. Red dotted box shows the optimal amplification conditions for successful discrimination of WT and N501Y samples. (**C**) Schematic for the double amplicon RAY. Positions of primers and sgRNAs for the N501Y sgRNA and WT S-gene sgRNA are shown. The two amplicons are highlighted in green and red. (**D**) Key steps in the assay and overall time is indicated. (**E**) Representative image showing the limit of detection of

*Figure 4 continued on next page*

*Figure 4 continued*

RAY for serial dilutions of patient samples (N501Y or WT as indicated). The corresponding TOPSE and *ORF1ab* Ct values are shown below. Right panel shows a quantification of the image intensity values (n = 2 RAY replicates per sample). (**F**) Graph showing the distribution of sequenced confirmed WT (cyan dot) and N501Y (red triangle) containing patient samples detected through RAY. Dotted line depicts the cut-off for N501Y sgRNA. (**G**) RAY outcomes on E484K and T716I mutations from patient samples. (**H**) Outcome of RAY showing minimal cross-reactivity of N501Y sgRNA on the E484K amplicon. (**I**) Increased signal intensity of N501Y RAY on patient samples upon using phosphorothioate modified synthetic sgRNA as compared to *in vitro* synthesized sgRNA.

multiple steps, low viral titres can be missed out during sequencing assays for variant calling. The RAY pipeline uses an amplification step followed by enzymatic binding and is reproducible across a wide range of viral titres tested.

The main advantage of sequencing COVID-19 samples is the generation of cumulative information about all mutations co-existing in a given variant (*Chiu and Miller, 2019* and *Vandenberg et al., 2021*). This information is valuable to track and trace how lineages evolve and has led to the identification of mutations in the first place. RAY can identify only known VOCs and is therefore useful as a diagnostic tool for the early detection of variant signatures in a sample. For example, determining whether a sample belongs to the South African lineage through E484K specific RAY can allow early quarantine of the patient and prevent the spreading of a highly transmissible variant. Similarly, identifying a WT sample rapidly through RAY negates the necessity of sequencing such a sample thereby reducing the cost of the associated sequencing exercise. Together, this can ease the burden of surveillance and ensure both rapid disease control measures as well as the identification of newer variants efficiently.

In the current format (lateral flow assay), RAY has limitations in multiplexing several VOC candidates in a single assay. Although longer amplicons can be used to identify multiple mutations that appear close to each other, we observed a drop in sensitivity probably due to reduced PCR efficiency for longer amplicons. Similarly, to increase the throughput of the assay, converting the read-out into a fluorescence-based mode might be beneficial.

Perhaps, the most imperative area of improvement for RAY is increasing the number of possible mutations that can be targeted. The necessity of an NGG PAM at a fixed position away from the position of the target mutation limits the total number of targets that RAY can identify. This can be significantly expanded by systematically exploring more sites in the sgRNA where a mutation weakens the enzyme:substrate interaction. Alternatively, engineering the enzyme to generate PAM-flexible versions can be considered.

Routine sequencing of patient samples for disease surveillance is fraught with issues such as cost, logistics, and turnaround times. A typical deep sequencing cost per sample is several orders of magnitude higher than RAY (which costs ~7 USD per sample, *Supplementary file 3*) and the minimum turnaround time for sequencing is 36 hr. Importantly, sequencing runs are generally done on pooled samples to save on cost, workforce, and machine time, all of which necessitate the advent of alternate solutions that can identify variants through a simple pipeline. RAY -on the contrary- is scalable to any number of samples at a given time.

Sequencing also generates a large amount of data and requires physical storage and high-performance computing methodologies to record, analyze and interpret the data (*Chiu and Miller, 2019* and *Vandenberg et al., 2021*). In addition, dedicated personnel with a substantial understanding of sequence analysis and programming skills are necessary for identifying variants within reads generated from a sample. In contrast, RAY uses a visual readout on a paper strip leading to a binary decision about the sample. The automated image acquisition and recording through a smartphone app allows operators with a minimum skill set to process samples. This has the advantage of rapid screening, particularly at the point of care settings.

In this study, we report the development of a CRISPR diagnostic platform for detecting the major VOCs in the SARS-CoV-2 genome. The outcomes of this platform can be extended to other pathogenic SNVs and can provide a robust solution for rapid variant calling in patient samples. Considering the rise and spread of mutated lineages of the virus in multiple countries, it is imperative to develop technologies that can quickly and accurately detect these variants. Since the first-generation vaccines against SARS-CoV-2 have been developed against the parent strain, careful assessment of the variant lineages and their response to the vaccine will become imperative during the course of

the pandemic (*Fontanet et al., 2021*; *Walensky et al., 2021*). Novel variants, particularly those correlated with adverse disease manifestation, need to be controlled so that these do not spread rampantly across populations. Testing and isolating them will thus continue to be at the forefront of disease control measures.

## Acknowledgements

We thank all members of Chakraborty and Maiti labs for helpful discussions and valuable insights. This study was funded by CSIR Sickle Cell Anemia Mission (HCP0008), TATA Steel CSR (SSP2001) and a Lady Tata Young Investigator award (GAP0198) to DC. The sequencing was funded by the CSIR (MLP-2005), Fondation Botnar (CLP-0031), Intel (CLP-0034) and IUSSTF (CLP-0033). We acknowledge Dr. Mohd. Faruq (CSIR IGIB) for help with isolation of RNA from the VTM of the RT-PCR positive samples. We thank Bala Pesala, Geethanjali Radhakrishnan and the entire team at ADIUVO Diagnostics for help with optimizing the image analysis application TOPSE. We also thank Sridhar Sivasubbu, Vinod Scaria and the IndiCoVGEN team for helpful discussions and support related to the work.

## Additional information

### Competing interests

Mohd Azhar: Mohd. Azhar is currently an employee of TATA Medical and Diagnostics. Debojyoti Chakraborty: A patent application has been filed in relation to this work. (0127NF2019). The other authors declare that no competing interests exist.

### Funding

| Funder | Grant reference number | Author |
| --- | --- | --- |
| University Grants Commission | Graduate student fellowship | Manoj Kumar |
| CSIR | Research Associateship | Sneha Gulati<br>Poorti Kathpalia |
| Indian Council of Medical Research | Graduate Student fellowship | Asgar H Ansari |
| CSIR | Graduate Student fellowship | Rhythm Phutela<br>Sundaram Acharya<br>Mohd Azhar<br>Jayaram Murthy |
| CSIR | MLP 2005 | Rajesh Pandey |
| Fondation Botnar | CLP-0031 | Rajesh Pandey |
| Intel Corporation | CLP-0034 | Rajesh Pandey |
| IUSSTF | CLP-0033 | Rajesh Pandey |
| CSIR | HCP0008 | Souvik Maiti<br>Debojyoti Chakraborty |
| Tata Steel | SSP 2001 | Debojyoti Chakraborty |
| Lady Tata Memorial Trust | GAP0198 | Debojyoti Chakraborty |

The funders had no role in study design, data collection and interpretation, or the decision to submit the work for publication.

### Author contributions

Manoj Kumar, Sneha Gulati, Conceptualization, Data curation, Formal analysis, Writing - review and editing; Asgar H Ansari, Resources, Formal analysis; Rhythm Phutela, Sundaram Acharya, Akshay Kanakan, Ranjeet Maurya, Janani Srinivasa Vasudevan, Aparna S, Data curation; Mohd Azhar, Investigation, Methodology; Jayaram Murthy, Investigation; Poorti Kathpalia, Resources; Rajesh Pandey, Formal analysis, Supervision; Souvik Maiti, Conceptualization, Funding acquisition, Methodology,

Project administration, Writing - review and editing; Debojyoti Chakraborty, Conceptualization, Formal analysis, Supervision, Funding acquisition, Investigation, Methodology, Writing - original draft, Project administration, Writing - review and editing

### Author ORCIDs
Manoj Kumar https://orcid.org/0000-0003-0772-1399
Janani Srinivasa Vasudevan http://orcid.org/0000-0002-3381-5228
Debojyoti Chakraborty https://orcid.org/0000-0003-1460-7594

### Ethics
Human subjects: The present study was approved by the Ethics Committee, Institute of Genomics and Integrative Biology, New Delhi (CSIR-IGIB/IHEC/2020-21/01).

### Decision letter and Author response
Decision letter https://doi.org/10.7554/eLife.67130.sa1
Author response https://doi.org/10.7554/eLife.67130.sa2

# Additional files

### Supplementary files
• Supplementary file 1. Mutations across emerging SARS-CoV-2 lineages showing the possibility of RAY based targetability of VOCs/VOIs.

• Supplementary file 2. RAY outcomes on patient samples (WT or N501Y). The cutoff for $S_{WT}$ is 6.7 and $S_{N501Y}$ is 10.2.

• Supplementary file 3. Price estimate of RAY consumables.

• Supplementary file 4. List of oligos used in this study.

• Transparent reporting form

### Data availability
Sequencing data associated with the manuscript have been deposited to GISAID with the following numbers: EPI_ISL_911542, EPI_ISL_911532, EPI_ISL_911543, EPI_ISL_911533, EPI_ISL_911544, EPI_ISL_911534, EPI_ISL_911545, EPI_ISL_911535, EPI_ISL_911546, EPI_ISL_911536, EPI_ISL_911547, EPI_ISL_911537, EPI_ISL_911538, EPI_ISL_911540, EPI_ISL_911541, EPI_ISL_911539.

The following datasets were generated:

| Author(s) | Year | Dataset title | Dataset URL | Database and Identifier |
|---|---|---|---|---|
| Pandey R, Kanakan A, Janani SV, Maurya R, Murali AS, Sahni S, Khan A, Chattopadhyay P, Devi P, Mehta P, Kumar A, Rawat N | 2021 | SARS CoV-2 sequencing data | https://www.epicov.org/epi3/entities/tmp/tmp_sd_2021_07_14_22_41_qw71ju_3lh56ce0a1a5/EPI_ISL_911542.fasta | GISAID, EPI_ISL_911542 |
| Pandey R, Kanakan A, Janani SV, Maurya R, Murali AS, Sahni S, Khan A, Chattopadhyay P, Devi P, Mehta P, Kumar A, Rawat N | 2021 | SARS CoV-2 sequencing data | https://www.epicov.org/epi3/entities/tmp/tmp_sd_2021_07_14_23_12_qw71ju_3pd56ce0931d/EPI_ISL_911532.fasta | GISAID, EPI_ISL_911532 |
| Pandey R, Kanakan A, Janani SV, Maurya R, Murali AS, Sahni S, Khan A, Chattopadhyay P, Devi P, Mehta P, | 2021 | SARS CoV-2 sequencing data | https://www.epicov.org/epi3/entities/tmp/tmp_sd_2021_07_14_23_13_qw71ju_3pgs6ce09282/EPI_ISL_911543.fasta | GISAID, EPI_ISL_911543 |

| | | | | | |
|---|---|---|---|---|---|
| Kumar A, Rawat N | | | | | |
| Pandey R, Kanakan A, Janani SV, Maurya R, Murali AS, Sahni S, Khan A, Chattopadhyay P, Devi P, Mehta P, Kumar A, Rawat N | 2021 | SARS CoV-2 sequencing data | https://www.epicov.org/epi3/entities/tmp/tmp_sd_2021_07_14_23_14_qw71ju_3pj26ce09266/EPI_ISL_911533.fasta | GISAID, EPI_ISL_911533 | |
| Pandey R, Kanakan A, Janani SV, Maurya R, Murali AS, Sahni S, Khan A, Chattopadhyay P, Devi P, Mehta P, Kumar A, Rawat N | 2021 | SARS CoV-2 sequencing data | https://www.epicov.org/epi3/entities/tmp/tmp_sd_2021_07_14_23_16_qw71ju_3pml6ce091cf/EPI_ISL_911544.fasta | GISAID, EPI_ISL_911544 | |
| Pandey R, Kanakan A, Janani SV, Maurya R, Murali AS, Sahni S, Khan A, Chattopadhyay P, Devi P, Mehta P, Kumar A, Rawat N | 2021 | SARS CoV-2 sequencing data | https://www.epicov.org/epi3/entities/tmp/tmp_sd_2021_07_14_23_16_qw71ju_3ppp6ce0916e/EPI_ISL_911534.fasta | GISAID, EPI_ISL_911534 | |
| Pandey R, Kanakan A, Janani SV, Maurya R, Murali AS, Sahni S, Khan A, Chattopadhyay P, Devi P, Mehta P, Kumar A, Rawat N | 2021 | SARS CoV-2 sequencing data | https://www.epicov.org/epi3/entities/tmp/tmp_sd_2021_07_14_23_17_qw71ju_3pym6ce0905a/EPI_ISL_911545.fasta | GISAID, EPI_ISL_911545 | |
| Pandey R, Kanakan A, Janani SV, Maurya R, Murali AS, Sahni S, Khan A, Chattopadhyay P, Devi P, Mehta P, Kumar A, Rawat N | 2021 | SARS CoV-2 sequencing data | https://www.epicov.org/epi3/entities/tmp/tmp_sd_2021_07_14_23_18_qw71ju_3q0b6ce0957b/EPI_ISL_911535.fasta | GISAID, EPI_ISL_911535 | |
| Pandey R, Kanakan A, Janani SV, Maurya R, Murali AS, Sahni S, Khan A, Chattopadhyay P, Devi P, Mehta P, Kumar A, Rawat N | 2021 | SARS CoV-2 sequencing data | https://www.epicov.org/epi3/entities/tmp/tmp_sd_2021_07_14_23_19_qw71ju_3q2n6ce09531/EPI_ISL_911546.fasta | GISAID, EPI_ISL_911546 | |
| Pandey R, Kanakan A, Janani SV, Maurya R, Murali AS, Sahni S, Khan A, Chattopadhyay P, Devi P, Mehta P, Kumar A, Rawat N | 2021 | SARS CoV-2 sequencing data | https://www.epicov.org/epi3/entities/tmp/tmp_sd_2021_07_14_23_19_qw71ju_3q6d6ce094bf/EPI_ISL_911536.fasta | GISAID, EPI_ISL_911536 | |
| Pandey R, Kanakan A, Janani SV, Maurya R, Murali AS, Sahni S, Khan A, Chattopadhyay P, Devi P, Mehta P, Kumar A, Rawat N | 2021 | SARS CoV-2 sequencing data | https://www.epicov.org/epi3/entities/tmp/tmp_sd_2021_07_14_23_22_qw71ju_3qf46ce08f1f/EPI_ISL_911547.fasta | GISAID, EPI_ISL_911547 | |
| Pandey R, Kanakan A, Janani SV, Maurya R, Murali AS, Sahni S, Khan A, Chattopadhyay P, Devi P, Mehta P, Kumar A, Rawat N | 2021 | SARS CoV-2 sequencing data | https://www.epicov.org/epi3/entities/tmp/tmp_sd_2021_07_14_23_23_qw71ju_3qim6ce08e89/EPI_ISL_911537.fasta | GISAID, EPI_ISL_911537 | |
| Pandey R, Kanakan A, Janani SV, Maurya R, Murali AS, Sahni S, Khan | 2021 | SARS CoV-2 sequencing data | https://www.epicov.org/epi3/entities/tmp/tmp_sd_2021_07_14_23_24_qw71ju_3qlo6ce08e2a/ | GISAID, EPI_ISL_911538 | |

| | | | EPI_ISL_911538.fasta | |
| --- | --- | --- | --- | --- |
| Pandey R, Kanakan A, Janani SV, Maurya R, Murali AS, Sahni S, Khan A, Chattopadhyay P, Devi P, Mehta P, Kumar A, Rawat N | 2021 | SARS CoV-2 sequencing data | https://www.epicov.org/epi3/entities/tmp/tmp_sd_2021_07_14_23_25_qw71ju_3qs16ce08d8f/EPI_ISL_911540.fasta | GISAID, EPI_ISL_911540 |
| Pandey R, Kanakan A, Janani SV, Maurya R, Murali AS, Sahni S, Khan A, Chattopadhyay P, Devi P, Mehta P, Kumar A, Rawat N | 2021 | SARS CoV-2 sequencing data | https://www.epicov.org/epi3/entities/tmp/tmp_sd_2021_07_14_23_26_qw71ju_3qu36ce08d4f/EPI_ISL_911541.fasta | GISAID, EPI_ISL_911541 |
| Pandey R, Kanakan A, Janani SV, Maurya R, Murali AS, Sahni S, Khan A, Chattopadhyay P, Devi P, Mehta P, Kumar A, Rawat N | 2021 | SARS CoV-2 sequencing data | https://www.epicov.org/epi3/entities/tmp/tmp_sd_2021_07_14_23_26_qw71ju_3qx16ce08cf4/EPI_ISL_911539.fasta | GISAID, EPI_ISL_911539 |

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

## Appendix 1

### RAY: a detection assay for major VOCs of SARS-CoV-2
Materials and reagents

- Micropipettes
- Plastic ware (including sterile filter tips, PCR and micro centrifuge tubes).
- Personal protective equipment (PPE)
- General laboratory materials and reagents, commonly used in molecular biology labs

### Equipment

- Microfuge/Centrifuge as necessary
- Thermal cycler
- Thermal bath (Heat block)

### General precautions

- Set up the reactions in designated areas for sample processing, RT-PCR and amplification.
- Handle all RT-PCR amplified products in separate post-amplification areas to prevent contamination from amplified products.
- Use filter tips to set up all reactions.
- Set up positive control reactions at the end.

NOTE! It is highly recommended to open tubes containing amplified PCR products from patient samples and positive controls, in a designated post amplification area, physically separated from the room where nucleic acid (RNA) extraction happens.

### Preparation of CRISPR-Cas9 crRNAs (to be performed prior to sample handling under RNase free condition)

| Reagents | Volume (µl) | Final concentration |
|---|---|---|
| Forward Oligo (100 µM) | 1.25 | 2.5 µM |
| Reverse Oligo (100 µM) | 1.25 | 2.5 µM |
| Total Volume (with nuclease free water) | upto 50 µl | |

Synthesis of *in vitro* transcribed (IVT) crRNAs targeting SAR-CoV-2 *S* gene and N501Y mutation

- To assemble equimolar ratio of Forward and Reverse oligos (refer table below) for each target:
- Heat the reaction mix at 95˚C for 5 min followed by slow cooling at room temperature for 15 min.
- Perform *in vitro* transcription using commercially available T7 Polymerase based IVT kit as per recommended protocol. A sample is given below (MEGAscript T7 Kit, ThermoFisher Scientific).

Assemble the following reaction components at room temperature:

| Reagents | Volume (µl) | Final concentration |
|---|---|---|

*Continued on next page*

*continued*

| Reagents | Volume (µl) | Final concentration |
|---|---|---|
| Nuclease free water | upto 20 | |
| ATP (75 mM) | 2 | 7.5 mM |
| GTP (75 mM) | 2 | 7.5 mM |
| CTP (75 mM) | 2 | 7.5 mM |
| UTP (75 mM) | 2 | 7.5 mM |
| 10X Reaction Buffer | 2 | 1X |
| Enzyme Mix | 2 | - |
| Annealed oligo duplex from step 1 | 5 | - |

- Incubate the reaction mix overnight at 37°C.
- Add 1 µl of Turbo DNAse in the reaction mix and incubate at 37°C for 30 min.
- Heat inactivate at 70°C for 10 min.
- Optional: RNA can be visualized on a 2% agarose gel to check for its integrity.
- Column based RNA clean-up as per the commercially provided protocols (such as NucAway Spin Columns, AM10070, ThermoFisher Scientific).

Generation of chimeric gRNAs (crRNA:tracrRNA-FAM).

| Reagents | Volume (µl) | Final concentration |
|---|---|---|
| IVT synthesized crRNA | - | 1 µM |
| FAM labeled tracrRNA | - | 1 µM |
| Annealing Buffer (100 mM NaCl, 50 mM Tris-Cl pH 8.0, 1 mM MgCl$_2$) | upto 50 | |

Heat the reaction mix at 95°C for 5 min followed by slow cooling at room temperature for 15 min.

NOTE!  The crRNA and chimeric gRNA products can be produced in bulk and stored at −20°C for long term use.

## FELUDA-based detection for COVID-19 (RT-PCR)

1. Extract patient RNA according to CDC recommendations.
2. Set-up single step RT and PCR reaction.

Assemble the reaction components as below (using Biotin-labeled primers):

| Reagents | Volume (µl) | Final concentration |
|---|---|---|
| Forward Biotinylated Primer (10 µM) | 0.2 | 200 nM |
| Reverse Biotinylated Primer (10 µM) | 0.2 | 200 nM |
| dNTPs (10 mM) | 0.1 | 100 µM |
| 10X Reaction Buffer (200 mM Tris-Cl pH 8.4, 500 mM KCl) | 1 | (1X) |
| MgCl$_2$ (50 mM) | 0.3 | 1.5 mM |
| DMSO (100%) | 0.3 | 3% |
| Taq DNA polymerase (5 U/µl) | 0.05 | 0.025 U/µl |
| Reverse Transcriptase (200 U/µl) | 0.25 | 5 U/µl |
| RNA sample (≥ 5 ng)* | As per sample | |
| RNase inhibitor (Optional, in casenot present with RT enzyme) 20 U/µl | 0.2 | 0.4 U/µl |
| Total Volume (with nuclease free water) | upto 10 | |

Optional: In order to confirm amplification after single step RT and PCR, a volume of 1–2 µl from NTC and PC can be checked on a 2% agarose gel.

NOTE! For a single set of reactions one non-template control (NTC) and one positive control (PC) should be included, to check for non-specific amplification.

## Reaction conditions

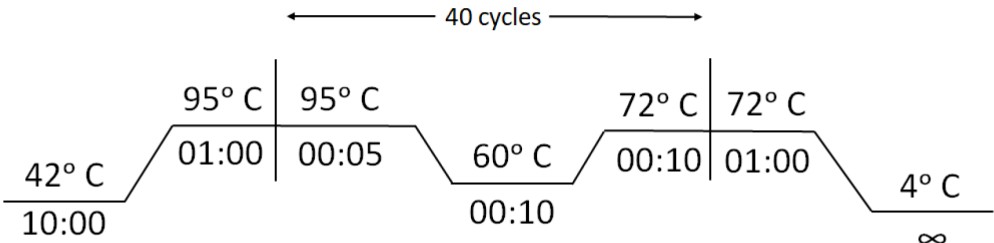

**Appendix 1—figure 1.** Cycling conditions for Reverse Transcription-PCR for RAY .

3. Prepare dFnCas9-chimeric gRNA-RNP complexes for the samples to be tested. RNP complex against SARS-CoV-2 S gene and N501Y mutation should be assembled for each sample. Incubate dFnCas9 protein with Chimeric FAM-labeled guide RNA to generate RNP complexes for 10 min at Room Temperature (RT).

| Reagents | Volume (µl) | Final concentration |
|---|---|---|
| dFnCas9 protein (1 µM) | 1.0 | 100 nM |
| Chimeric FAM-labeled gRNA (1 µM) | 1.0 | 100 nM |
| Total Volume (with Buffer containing 20 mM HEPES pH 7.5, 150mM KCl, 10% glycerol, 1mM DTT and 10 mM MgCl$_2$) | upto 5 | |

4. Add 5 µL of the biotinylated amplicon (from Step 2) to 5 µL of the dFnCas9 RNP complex (from Step 3).

NOTE! For a single set of reactions one non-template control (NTC) and one positive control (PC) should be included, to check dFnCas9 RNP specificity.

5. Incubate the reaction mix (containing RNP complex and amplicon substrate) at 37˚C for 10 min in aheating block or water bath.
6. Add 80 µL of dipstick buffer (provided) to each tube containing 10 µL of the reaction mix from the previous step.
7. Insert Milenia HybriDetect one lateral flow strip directly into reaction tubes.
8. Allow the solution to migrate into the strip for 2 min at room temperature and observe the result.
Notes:

- NTC (non-template control) to be used as negative control.
- For accurate results, positive samples show up in the test band within 2 min while negative samples show very weak or no signal.
- Incubating for longer times leads to increasing background signal intensity at test band location making interpretation ambiguous.
- Signals between positive and negative assays can also be interpreted by densitometry analysis from images captured using any photographic device.

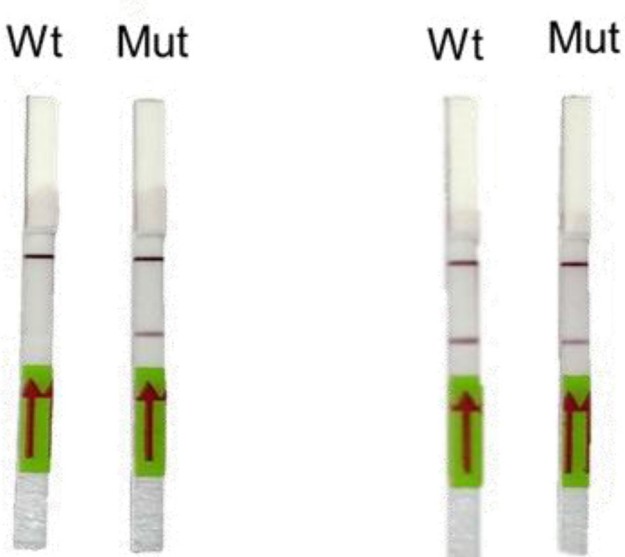

**Appendix 1—figure 2.** Representative image of RAY outcomes on a paper strip for sample having the N501Y mutation (Mut) or the parent CoV-2 sequence (Wt). Left panel shows outcome with $S_{N501Y}$ sgRNA and the right panel shows the outcome with $S_{WT}$ sgRNA.

## Image analysis through TOPSE smartphone app:

1. T.O.P.S.E (True Outcome Predicted via Strip Evaluation), starts by clicking 'GET STARTED' on the home page, which further directs to an app page where access to previous recorded data is given through entering a operator ID (eg. XYZ) or a new session of data recording begins. The app then further asks to record a background value for No template control (NTC).
2. Upon selecting 'NEW NTC', a new app page having a strip holder pops up, once the strip is placed in the holder, a numerical band intensity for the test band is calculated to set a background value. After clicking on 'DONE', a pop up asks to record for a patient sample.
3. In the next page, clicking on 'OPEN CAMERA', takes the operator to another strip holder to record test band intensity of the patient sample strip. If the calculated test band intensity is above NTC threshold, the sample will be called as 'POSITIVE'.
4. Once a recording session is done, one can record for another sample just by clicking onto 'ADD PATIENT', the app will automatically use the previous NTC threshold value and need to be set again. For eg. if the calculated test band intensity for the second sample is below NTC threshold or within the error limit of the app, the sample will be called as 'NEGATIVE'.
5. For using TOPSE to perform N501Y RAY assay, the test intensity of the sample needs to be >10.2 for a sample to be called positive for the N501Y assay. The S gene intensity has to be >6.7 for the sample to be called positive for SARS CoV-2.
6. Strips may be discarded according to standard procedures.

## List of primers and oligos are mentioned in *Supplementary file 4*
### Reagents used

MEGAscript T7 Transcription Kit (Cat. No. AM1334, ThermoFisher Scientific), Quantitect RT (Cat. No. 205311, Qiagen), Taq DNA Polymerase (Cat. No. 18038018, ThermoFisher Scientific), 5' end biotinylated primers (Merck, Darmstadt, Germany), 3' end 6 – Fluorescein amidite (6-FAM) FnCas9 tracrRNA (GenScript Biotech (New Jersey, USA)) or Merck (Darmstadt, Germany), MILENIA HybriDetect1 (Cat. No. MILLENIA 01, TwistDx, UK or Giessen, Germany).

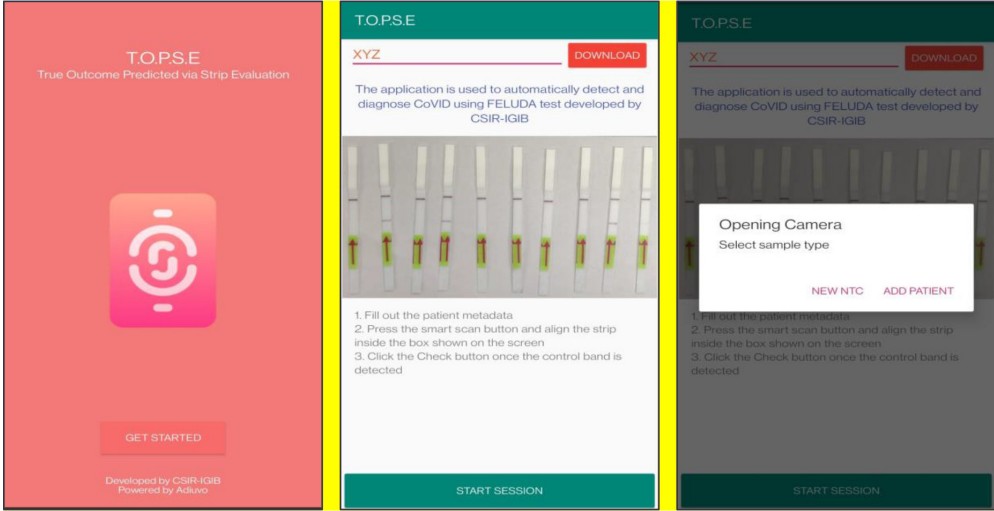

**Appendix 1—figure 3.** Representative screen-shots showing TOPSE app home page and user interface.

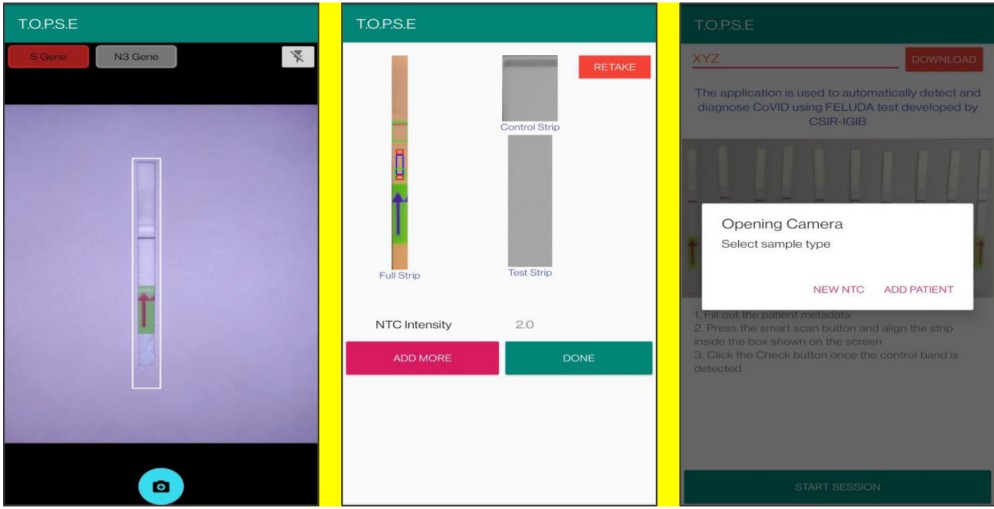

**Appendix 1—figure 4.** Representative screen-shots showing TOPSE image acquisition for NTC background correction.

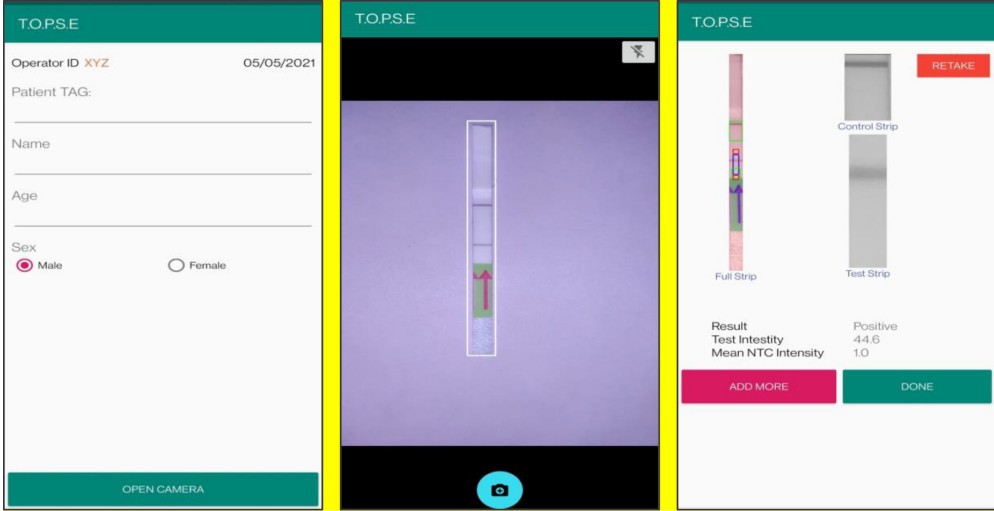

**Appendix 1—figure 5.** Representative screen-shots showing TOPSE data entry step and image acquisition for a sample identified as ''Positive''.

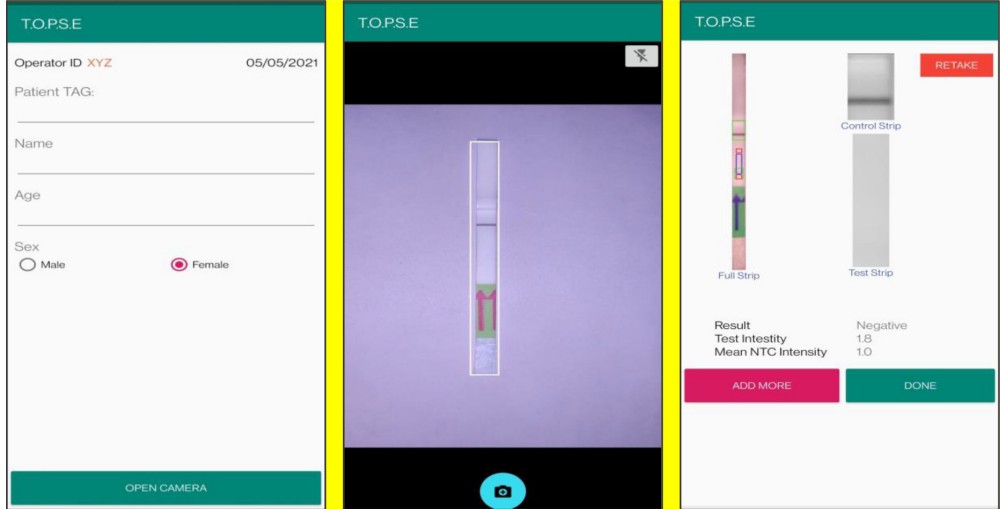

**Appendix 1—figure 6.** Representative screen-shots showing TOPSE data entry step and image acquisition for a sample identified as ''Negative''.

