## [Decision Letter]

**Acceptance summary:**

This paper is likely to be of interest to those interested in developing diagnostic tools for rapid low cost detection of single nucleotide variants in viral and host pathogenic sequences; genetic disorders and development of precision medicine. Although the key scientific claims are an extension to a recently developed CRISPR diagnostic platform by the same group, this advance has potential application to the COVID-19 pandemic.

The manuscript was revised to provide data and detailed methodologies to support the detection of two major variants of SARs CoV-2. It has been re-written such that the reader need not be familiar with the previous methodology published by the authors, which has been extended to detect these variants.

**Decision letter after peer review:**

Thank you for submitting your work entitled "RAY: CRISPR diagnostic for rapid and accurate detection of SARS-CoV-2 variants on a paper strip" for consideration by *eLife*. Your article has been reviewed by 3 peer reviewers, and the evaluation has been overseen by a Reviewing Editor and K VijayRaghavan as the Senior Editor.

The manuscript has some issues that need to be addressed, as outlined below:

This paper will be of interest to scientists interested in developing diagnostic tools for rapid low cost detection of SNVs in viral and host pathogenic sequences; genetic disorders and development of precision medicine. Although this tool has immediate application in healthcare during the current pandemic, the key scientific claims of the manuscript are an extension to a recently developed CRISPR diagnostic platform by the same groups. Further data and detailed methodologies need to be provided to support generalizing of the conclusions for applications to a broader field beyond detection of the two 2 variants of SARs-CoV-2.

Briefly, we request a revised manuscript. It is written such that the reader must already be familiar with a previous paper published by the authors, of which this is an extension to detect variants. The manuscript should be stand alone and include all information about this assay and the method validation result. As the title of this manuscript is "detection of variants on a paper strip", generalizability must be addressed if this assay is able to do that. If not, the authors must discuss how to achieve generality and revise the title to "detection of common SARS-CoV-2 variants". In this scenario, the results can support the title.

*Reviewer 1:*

Researchers from different groups have employed CRISPR-Cas9 protein variants (Cas13, and Cas12a) to develop nucleic acids detection platforms such as SHERLOCK (2017, Science, 356, 438; 2018, Science, 360, 439) and DETECTR (2018, Science, 360, 436). Both SHERLOCK and DETECTR rely on the "collateral effect" – promiscuous cleavage and degradation of neighboring ssRNA and ssDNA by Cas13 and Cas12a. Upon target binding, it will cleave the ssDNA probe. These CRISPR-based detections were used for SARS-CoV-2 detection after then (2020, Nature Biotechnology, 38, 870).

Authors previously reported that Francisella novicida (FnCas9) could be used for gene editing. In authors' preprint project, they reported the high specificity of FnCas9 to the point mismatch at the specific position of gRNA and there is no "collateral effect" of FnCas9. Based on this discovery, they developed a non-cleavage, affinity-based method for the nucleobase detection with single nucleotide mismatch sensitivity (ref 25).

Authors leverage the single nucleotide mismatch sensitivity of FnCAS9 and report herein a non-cleavage, affinity-based SARS-CoV-2 variants detection.

Strengths:

– This assay provides a simple, rapid way to detect SARS-CoV-2 and its SNV.

– Authors demonstrated the detection of SARS-CoV-2 and SAR-S-CoV2 variants (N501Y) in 18 patient samples and validated the samples by sequencing.

Weaknesses:

– Although authors cite their previous preprint paper (similar assay called "FELUDA"), authors didn't describe any sensing mechanism of this assay (RAY). Readers cannot understand this assay by reading this manuscript only. The lack of description in this manuscript strongly affects readers to understand the experiments. Authors assume readers have read another article from the same group and understand the design and mechanism of their "FELUDA" assay. For example, Authors didn't mention that active FnCas9 was used in the experiment in Figure 2a while catalytic inactive FnCas9 was used for SNV detection assay "RAY".

– Only N501Y detection was demonstrated in this project. Authors did not demonstrate the generalizability of this assay by demonstration of detecting other SARS-CoV-2 variants. Furthermore, authors mentioned that only 10/32 SNV sites were targetable by FnCas9.

– The concept / sensing mechanism of this assay is very similar to the assay reported in the same group (ref 25). They used a lot of results / discovery in the preprint (ref 25) to develop this assay and there are only 4 sets of new data reported in this manuscript.

There are several points that may help authors to improve this project.

"We found that out of the 32 defining SNVs 100 reported across all the three lineages, 10 sites were targetable by FnCas9 (Figure 1B,C, Supplementary Table 1)"

Figure 1B shows the evolution of SARS-CoV-2 virus. How is it related to this statement? Figure 1C shows the position of different mutations in the genome. Why is it important to show the position of the mutations? Is it related to the targetability of FnCas9? Explanation is needed.

Authors should explain how they find 10 of 32 single nucleotide variation (SNV) sites are targetable by FnCas9. Why only 10/32 SNV sites are targetable? Authors should explain this result. Without this explanation, readers will not understand how this assay works and how to use this assay to detect other SNV.

As only 10 of 32 SNV sites are targetable, authors should discuss the generalizability of this assay.

"Thirdly, we made modifications at the level of amplicon length, PCR conditions, and RNP concentration to generate the most optimal protocol that can produce a clear discernible signal on a commercially available paper strip"

No data is shown about PCR conditions and RNP concentration optimization. Why does the assay not work well when the amplicon length is 541 bp or less?

In my view, the combination of the preprint paper (ref 25) and this manuscript will make it a detailed, comprehensive project.

There are additional points that need addressing to improve this paper.

Only N501Y detection was demonstrated in this paper strip assay. Paper strip detection of other targetable SNV sites could show the generalizability of this assay in specific conditions.

This manuscript has a lot of abbreviations, but some of them only appear 1 or 2 times. It makes this manuscript difficult for readers to read.

It is confused that Wt, mut and N501Y are all present in Figure 2a. Authors are suggested to label it more clearly to show that it is WT and mutated sgRNA (N501Y).

"Finally, we tested RAY on RNA extracted from samples of RT-PCR positive SARS-CoV-2 infected individuals who returned to India from the United Kingdom between December 2020 and January 2021."

The travel history of the patient is not necessary for assay validation.

*Reviewer 2:*

The authors report the development of a CRISPR diagnostic platform for detecting variants (N501Y and T716I) in SARS-CoV-2. This is based on a technology, recently developed by the same research groups, using a Cas9 ortholog from Francisella novicida (FnCas9) based enzymatic readout for detecting nucleotide sequences. RAY (Rapid variant Assay), detect both the infection as well as the presence of the common N501Y mutation present in the 3 new variants and distinguish it from the parent CoV-2 lineage. The authors also adapt RAY to a lateral flow paper strip assay for low-cost rapid detection. This represents a novel sequencing-free rapid detection system for CoV-2 mutations.

Although the paper does have major strengths in principle, the authors have not provided sufficient data and information directly for an end user to adapt this platform to their use. For a stand-alone resource article, authors must provide all the critical information within the text, figures, methods and supplementary figures required to understand the range, specificity, sensitivity, limits and error prone aspects (if any) of the technology.

Further information and clarification are required for the following points:

Primer design and selection:

1. Given that the authors mention that they successfully established that FnCas9 is unable to bind or cleave targets having two mismatches -in the 2nd and 6th position; why were PAMs searched with mismatches in the 16th and 19th position (Figure 1D)?

2. How many pairs of flanking primers need to be designed for a specific point mutation? What experimental output would demonstrate optimal primer design?

3. More details on the amplicons are required.

IVC Detection

1. What size fragments were generated? Was only one sgRNA used for each N501Y and T716I?

In figure 2a it seems like the amount of wild type amplicon is higher or equal to one of the cleaved fragments in the N501Y sample gel (mut lane). Is this due to poor cleavage by the FnCas9? A densitometry analysis would provide better quantification of fragment formations.

2. What is the specificity, selectivity for each of these variants? Do altering sgRNA change the specificity? What conditions are necessary for the reaction to go to completion? How does one account for false negatives; is there a low copy number limit for this detection system. No experimental results have been provided for any of this.

3. For the reaction between FnCas9 and sample/amplicons what are the ratio ranges; concentrations; volumes to be followed?

LFA Detection

1. Please elaborate the protocol for "modified single step" RT-PCR.

2. The authors mention that they eliminate one of the biotin label primers to reduce the chances of background signal due to non-specific interactions of the unused biotin primers with the streptavidin. The authors should compare and quantitate the output using various methods used in the current work as well as previous FELUDA paper [double biotin labels; synthetic phosphothioate modified RNA etc]

3. Authors should explain the difference between FnCas9 and dFnCas9 in at least the methods.

4. What are the limits of detection of RAY? Is it similar to FELUDA? How does changing sgRNA/ chimeric gRNA effect the detection limit? No experimental results have been provided for this.

5. If the only method of detection analysis is TOPSE then the authors must provide more details of how to access/use the app.

*Reviewer 3:*

The work by Kumar et al. presents a new diagnostic technique (called "RAY") for detecting SARS-CoV-2 virus infection (responsible for the current COVID-19 pandemic). The researchers use a CRISPR-based method to provide a "quick" assay that can distinguish COVID positive and negative samples. This method is an extension of their previous work on a COVID-19 assay called FELUDA (https://www.medrxiv.org/content/10.1101/2020.09.13.20193581v1). The focus of the current work is to specifically identify COVID-positive samples as well as the particular variant of the viral genome. Typically, the different SARS-CoV-2 variants could differ by a few differences in the genome sequence. This work uses the specificity of the CRISPR reaction: the enzymatic cleavage only occurs for a completely complementary target and not for a target with mutations. The authors developed a lateral flow assay, where the signal corresponding to detection of the viral RNA and/or the mutant is read out as a line on a paper strip (similar to pregnancy strips). Appearance of these lines indicate presence of the virus in the sample. The work used 8 COVID positive samples, 8 positive samples with the mutation N501Y (earlier shown to be present in the British, South African and Brazilian variants of SARS-CoV-2), and 12 COVID negative samples, and identified all the clinical samples accurately.

In terms of the scope, the current work demonstrates COVID19 mutation detection with an easy lateral flow-based readout, within ~75 minutes. While the assay has been shown to be able to detect 20 samples accurately, a larger data set might be needed to evaluate the sensitivity of the assay and the range of viral loads it can detect. The past year has seen a number of new techniques for COVID19 diagnosis, all of which will be useful for the still ongoing large-scale testing worldwide. In that context, this assay has practical and immediate use for COVID19 detection if it can be commercialized, especially with the rise of a number of variants of the SARS-CoV-2 virus.

I liked reading the paper – it was very clear and crisp, and the data was well presented in both the text and the figures. However, as written, the paper seems to rely heavily on the reader's knowledge of their earlier work on FELUDA. From a reader's point of view, I felt that certain aspects of the assay, methodology and scope of immediate use could be expanded here even if it is a repeat from the authors' FELUDA work, to make this a stand-alone paper.

Discussion can be included on the following points. While these are not technical concerns, I think it will be informative for scientific and lay audience to know where the assay stands in terms of immediate impact for COVID19 pandemic.

– Metrics of the assay such as sensitivity and specificity (even if referring to previous works).

– Scope of the assay: Do the authors intend to use this as a clinical test or at POC? Specific aspects can be discussed for either or both of these scenarios.

– For the assay to be useful in POC setting, sample processing is an important step. Assuming samples are collected on swabs, RNA needs to be extracted before RAY can be performed. How can this be brought into the regular workflow for the assay to work at a testing site outside a lab or clinical facility? How will this affect the cost of the assay? The tally of the assay cost excludes RNA extraction. How can such an enzymatic approach be performed in a remote site without a cold chain?

---

## [Author Response]

Briefly, we request a revised manuscript. It is written such that the reader must already be familiar with a previous paper published by the authors, of which this is an extension to detect variants. The manuscript should be stand alone and include all information about this assay and the method validation result. As the title of this manuscript is "detection of variants on a paper strip", generalizability must be addressed if this assay is able to do that. If not, the authors must discuss how to achieve generality and revise the title to "detection of common SARS-CoV-2 variants". In this scenario, the results can support the title.

We thank the Senior and Reviewing Editors of *eLife* for the positive and helpful suggestions and for considering that this tool has immediate application in healthcare during the current pandemic. We are now submitting a revised manuscript taking care of the points raised by the reviewers, both in terms of a better description of the technology and its modifications and improvements alongside commenting on the generalizability of the assay in the Discussion section. We accept the suggestion made about the title and have amended the same in the new version.

Reviewer 1:Researchers from different groups have employed CRISPR-Cas9 protein variants (Cas13, and Cas12a) to develop nucleic acids detection platforms such as SHERLOCK (2017, Science, 356, 438; 2018, Science, 360, 439) and DETECTR (2018, Science, 360, 436). Both SHERLOCK and DETECTR rely on the "collateral effect" – promiscuous cleavage and degradation of neighboring ssRNA and ssDNA by Cas13 and Cas12a. Upon target binding, it will cleave the ssDNA probe. These CRISPR-based detections were used for SARS-CoV-2 detection after then (2020, Nature Biotechnology, 38, 870).Authors previously reported that Francisella novicida (FnCas9) could be used for gene editing. In authors' preprint project, they reported the high specificity of FnCas9 to the point mismatch at the specific position of gRNA and there is no "collateral effect" of FnCas9. Based on this discovery, they developed a non-cleavage, affinity-based method for the nucleobase detection with single nucleotide mismatch sensitivity (ref 25).Authors leverage the single nucleotide mismatch sensitivity of FnCAS9 and report herein a non-cleavage, affinity-based SARS-CoV-2 variants detection.Strengths:– This assay provides a simple, rapid way to detect SARS-CoV-2 and its SNV.– Authors demonstrated the detection of SARS-CoV-2 and SARS-CoV-2 variants (N501Y) in 18 patient samples and validated the samples by sequencing.Weaknesses:– Although authors cite their previous preprint paper (similar assay called "FELUDA"), authors didn't describe any sensing mechanism of this assay (RAY). Readers cannot understand this assay by reading this manuscript only. The lack of description in this manuscript strongly affects readers to understand the experiments. Authors assume readers have read another article from the same group and understand the design and mechanism of their "FELUDA" assay. For example, Authors didn't mention that active FnCas9 was used in the experiment in Figure 2a while catalytic inactive FnCas9 was used for SNV detection assay "RAY".

We thank Reviewer 1 for his/her assessment of the assay and appreciating its ability to detect SNVs that constitute the VOCs in the SARS-CoV-2 genome. We understand the concerns raised by the Reviewer about the description of the sensing mechanism of RAY and the lack of clarity about the active/inactive versions of the FnCas9 used in Figure 2a. In the revised version we have made changes reflecting a detailed explanation of the assay and its modifications and included a comprehensive Supplementary Note describing the assay implementation.

– Only N501Y detection was demonstrated in this project. Authors did not demonstrate the generalizability of this assay by demonstration of detecting other SARS-CoV-2 variants. Furthermore, authors mentioned that only 10/32 SNV sites were targetable by FnCas9.

We thank Reviewer 1 for this point. Although we achieved the design of a RAY strategy for multiple variants, we couldn't validate them all due to time and sample availability constraints (as not all the variants were detected in the samples that were sequenced). However, we picked three of the reported VOCs in the samples we had access to (namely N501Y, E484K, and T716I) and validated RAY on patient RNA samples. We were able to detect all three variants using our assay successfully. We chose to perform intensive validation and standardization with N501Y over a larger sample set owing to its common occurrence across multiple lineages (Figure 4).

The Reviewer has rightly mentioned about 10/32 SNV sites that were targetable by FnCas9 in our original submission. Due to the rapid progress and evolution of the SARS-CoV-2 virus, currently (as of May 2021), 13 emerging lineages have been reported (Figure 1B, revised manuscript). Out of these, 5 lineages have been designated as Variants of Concern (VOCs) and 8 lineages as Variants of Interest (VOIs) by the Centers for Disease Control and Prevention (https://www.cdc.gov/coronavirus/2019-ncov/cases-updates/variant-surveillance/variant-info.html#Interest). We reanalyzed the emerging lineages and found that there are 48 SNVs associated with these VOCs and VOIs and RAY design based on existing parameters can target 19 of these SNVs. Importantly, RAY is able to detect every VOC/VOI through at least one SNV. We represent this data in Supplementary Table 1 and Figure 1 (revised manuscript).

Interestingly, N501Y is associated with 3/5 VOCs (B.1.1.7, P.1, and B.1.351). Thus, by detecting this variant, RAY can detect whether a sample belongs to one of these VOCs. In the remaining VOCs (B.1.427 and B.1.429) RAY for the SNV W152C SNV can be used in detection. However, this has not been practically validated.

– The concept / sensing mechanism of this assay is very similar to the assay reported in the same group (ref 25). They used a lot of results / discovery in the preprint (ref 25) to develop this assay and there are only 4 sets of new data reported in this manuscript.

We thank the Reviewer for this point but would politely disagree. Although the sensing mechanism occurs through FnCas9 and as was described before (Azhar, Phutela, Kumar, Ansari, et al. *Biosensors and Bioelectronics,* 2021), the current manuscript shows the detection of SARS-CoV-2 VOCs on a paper strip, which in our opinion, is both novel and has a high translational impact. Currently, identifying SARS-CoV-2 variants from patient samples exclusively relies on deep sequencing that is both costly and time-consuming. Thus, the value of this manuscript goes beyond simple discovery to a more useful public resource methodology which we hope gets implemented on a larger scale. To enable this paper-strip readout that is robust and performs with high sensitivity and specificity required several rounds of iteration and modification of the original FELUDA protocol and cannot be seen as an extension of the previous work.

There are several points that may help authors to improve this project."We found that out of the 32 defining SNVs 100 reported across all the three lineages, 10 sites were targetable by FnCas9 (Figure 1B,C, Supplementary Table 1)"Figure 1B shows the evolution of SARS-CoV-2 virus. How is it related to this statement? Figure 1C shows the position of different mutations in the genome. Why is it important to show the position of the mutations? Is it related to the targetability of FnCas9? Explanation is needed.

We thank the Reviewer for pointing this out. We agree that Figure 1B does not appropriately correlate with the statement made and has now been replaced in the revised manuscript. We have also explained this better in the manuscript and hope that these modifications will help understand the targeting scope of RAY.

Authors should explain how they find 10 of 32 single nucleotide variation (SNV) sites are targetable by FnCas9. Why only 10/32 SNV sites are targetable? Authors should explain this result. Without this explanation, readers will not understand how this assay works and how to use this assay to detect other SNV.As only 10 of 32 SNV sites are targetable, authors should discuss the generalizability of this assay.

We thank the Reviewer for this point. The mechanism by which FnCas9 based identification of SNVs has now been explained in detail in the revised manuscript (Figure 1 and Methods). We have also included a statement about the generalizability of this assay in the discussions section and made an amendment to the manuscript title to reflect this.

"Thirdly, we made modifications at the level of amplicon length, PCR conditions, and RNP concentration to generate the most optimal protocol that can produce a clear discernible signal on a commercially available paper strip"No data is shown about PCR conditions and RNP concentration optimization. Why does the assay not work well when the amplicon length is 541 bp or less?In my view, the combination of the preprint paper (ref 25) and this manuscript will make it a detailed, comprehensive project.

We thank the Reviewer for pointing this out. In the revised manuscript, we have demonstrated the conditions for optimization in Figure 3 and Figure 4. We have also discussed the importance of amplicon length on RAY performance in the manuscript and included a detailed protocol as Supplementary Note 1.

There are additional points that need addressing to improve this paper.Only N501Y detection was demonstrated in this paper strip assay. Paper strip detection of other targetable SNV sites could show the generalizability of this assay in specific conditions.

We thank the reviewer for pointing this out. We have now demonstrated the detection of 2 more SNVs (E484K and T716I) in addition to N501Y on paper strips to show that several SNVs can be targeted through RAY (Figure 4G-H, revised manuscript).

This manuscript has a lot of abbreviations, but some of them only appear 1 or 2 times. It makes this manuscript difficult for readers to read.It is confused that Wt, mut and N501Y are all present in Figure 2a. Authors are suggested to label it more clearly to show that it is WT and mutated sgRNA (N501Y)."Finally, we tested RAY on RNA extracted from samples of RT-PCR positive SARS-CoV-2 infected individuals who returned to India from the United Kingdom between December 2020 and January 2021."The travel history of the patient is not necessary for assay validation.

We thank the reviewer for mentioning these points. In the revised manuscript, we have taken care to refer to abbreviations judiciously and have removed the travel history information from the main manuscript text as suggested.

Reviewer 2:The authors report the development of a CRISPR diagnostic platform for detecting variants (N501Y and T716I) in SARS-CoV-2. This is based on a technology, recently developed by the same research groups, using a Cas9 ortholog from Francisella novicida (FnCas9) based enzymatic readout for detecting nucleotide sequences. RAY (Rapid variant Assay), detect both the infection as well as the presence of the common N501Y mutation present in the 3 new variants and distinguish it from the parent CoV-2 lineage. The authors also adapt RAY to a lateral flow paper strip assay for low-cost rapid detection. This represents a novel sequencing-free rapid detection system for CoV2 mutations.Although the paper does have major strengths in principle, the authors have not provided sufficient data and information directly for an end user to adapt this platform to their use. For a stand-alone resource article, authors must provide all the critical information within the text, figures, methods and supplementary figures required to understand the range, specificity, sensitivity, limits and error prone aspects (if any) of the technology.

We are grateful to Reviewer 2 for appreciating the technology and critically evaluating the work. We value the concerns raised for providing detailed information within the text, figures, methods, and supplementary figures for better implementation of the assay by an end-user. We have now addressed each of these points in detail in the revised manuscript.

Further information and clarification are required for the following points:Primer design and selection:1. Given that the authors mention that they successfully established that FnCas9 is unable to bind or cleave targets having two mismatches -in the 2nd and 6th position; why were PAMs searched with mismatches in the 16th and 19th position (Figure 1D)?2. How many pairs of flanking primers need to be designed for a specific point mutation? What experimental output would demonstrate optimal primer design?3. More details on the amplicons are required.

We thank Reviewer 2 for pointing this out. We have addressed these as follows:

1. The PAM search from the 16th and 19th positions was done based on our observation that the combination of these positions could also allow the FnCas9 to discriminate between its targets. In the revised manuscript, we have shown the *in vitro* cleavage outcomes for both 2 and 6 and 16 and 19 mismatched positions (Figure 1A). We apologize for not including this crucial piece of data in the original version.

2. A single primer pair of flanking primers are needed for a specific point mutation. The experimental output that demonstrates optimal primer design is the distinct difference in band intensities between the WT and SNV samples. In the revised manuscript, we have explained the methodology for smartphone-based evaluation of band intensities and subsequently calling the samples WT or Mutant.

3. We have provided details of the amplicon design in the revised manuscript.

IVC Detection1. What size fragments were generated? Was only one sgRNA used for each N501Y and T716I?In figure 2a it seems like the amount of wild type amplicon is higher or equal to one of the cleaved fragments in the N501Y sample gel (mut lane). Is this due to poor cleavage by the FnCas9? A densitometry analysis would provide better quantification of fragment formations.

We have indicated the size of the fragments in the revised manuscript (Figure 2A). Two different sgRNAs were used for N501Y and T716I correspondingly. The sequences of these sgRNAs are provided in Supplementary Table 4 in the revised manuscript. Densitometric analysis showed 72% cleavage of the mutant substrate and 3% cleavage of the WT substrate for N501Y and 83% and 1% respectively for T716I. Taken together this shows that the mismatch containing sgRNAs are highly specific in distinguishing the two substrates. In the revised manuscript, we have shown the outcome of T716I only as a lateral flow assay for clarity (Figure 4G).

2. What is the specificity, selectivity for each of these variants? Do altering sgRNA change the specificity? What conditions are necessary for the reaction to go to completion? How does one account for false negatives; is there a low copy number limit for this detection system. No experimental results have been provided for any of this.3. For the reaction between FnCas9 and sample/amplicons what are the ratio ranges; concentrations; volumes to be followed?

The specificity and selectivity of these variants are high. In the revised manuscript, we present data from 37 WT and 22 N501Y samples and observe a sensitivity of 86 % and specificity of 97 % on LFA (Figure 4 in the revised manuscript). Since the discrimination of SNVs through RAY follows a design based on introducing mismatches, the possibility of alteration in sgRNA doesn’t arise since the mismatch positions are fixed with reference to the mutation.

For both the PCR and the subsequent CRISPR reaction, the presence of components (primers, enzymes, buffers, etc.) are in excess (details in Supplementary Note 1), and therefore the reaction proceeds to completion. In our previous study (Azhar, Phutela, Kumar, Ansari, et al. *Biosensors and Bioelectronics,* 2021), we had optimized the time required for the FnCas9 based substrate-binding reaction to proceed to completion. In this manuscript, we have adapted the same reaction time for RAY.

As seen in the revised manuscript (Figure 4E, F), false-negative results arise when the test band intensity is not high enough for visual/app-based detection. This can happen when the viral load is extremely low or the PCR efficiency is not optimal. Alternatively, a very low-intensity band can also be due to degradation of the CRISPR components in the assay (dFnCas protein or sgRNA).

The limit of detection expressed as a corresponding RT-PCR Ct value has now been presented in the revised manuscript (Figure 4E).

LFA Detection1. Please elaborate the protocol for "modified single step" RT-PCR.

We provide a detailed protocol of the modified PCR and LFA steps in Supplementary Note 1.

2. The authors mention that they eliminate one of the biotin label primers to reduce the chances of background signal due to non-specific interactions of the unused biotin primers with the streptavidin. The authors should compare and quantitate the output using various methods used in the current work as well as previous FELUDA paper [double biotin labels; synthetic phosphothioate modified RNA etc]

We thank Reviewer 2 for pointing this out. We have now addressed all the points raised (single vs. double biotin labels) in Figure 4 of the revised manuscript. In our earlier study, we found that phosphorothioate modified synthetic sgRNAs have better stability as compared to *in vitro* synthesized sgRNAs over multiple rounds of freeze-thaw. Indeed, the same was observed when we performed RAY with phosphorothioate modified N501Y sgRNAs (Figure 4I).

3. Authors should explain the difference between FnCas9 and dFnCas9 in at least the methods.

We express regret for not clarifying this point well in the initial submission. We have now explained the difference clearly in the revised manuscript.

4. What are the limits of detection of RAY? Is it similar to FELUDA? How does changing sgRNA/ chimeric gRNA effect the detection limit? No experimental results have been provided for this.

We thank Reviewer 2 for raising this point. We would like to explain that RAY is an assay for discriminating between two samples (WT and mutant strains of the virus) while FELUDA is for detecting the viral RNA in a single patient sample. Thus for FELUDA, direct quantification of the limit of detection was possible (up to a Ct value of 38, *Biosensors and Bioelectronics*, 2021). For RAY, we performed serial dilutions of the WT and mutant strains (N501Y) and found that it was able to discriminate the two accurately up to a Ct value of 34. We have presented this data in Figure 4E (revised manuscript).

5. If the only method of detection analysis is TOPSE then the authors must provide more details of how to access/use the app.

We thank the reviewer for mentioning this. We have now described the TOPSE based acquisition and final interpretation in Supplementary Note 1. The application has been uploaded to a web server, and details about accessing the app have also been included.

Reviewer 3:The work by Kumar et al. presents a new diagnostic technique (called "RAY") for detecting SARS-CoV-2 virus infection (responsible for the current COVID-19 pandemic). The researchers use a CRISPR-based method to provide a "quick" assay that can distinguish COVID positive and negative samples. This method is an extension of their previous work on a COVID-19 assay called FELUDA (https://www.medrxiv.org/content/10.1101/2020.09.13.20193581v1). The focus of the current work is to specifically identify COVID-positive samples as well as the particular variant of the viral genome. Typically, the different SARS-CoV-2 variants could differ by a few differences in the genome sequence. This work uses the specificity of the CRISPR reaction: the enzymatic cleavage only occurs for a completely complementary target and not for a target with mutations. The authors developed a lateral flow assay, where the signal corresponding to detection of the viral RNA and/or the mutant is read out as a line on a paper strip (similar to pregnancy strips). Appearance of these lines indicate presence of the virus in the sample. The work used 8 COVID positive samples, 8 positive samples with the mutation N501Y (earlier shown to be present in the British, South African and Brazilian variants of SARS-CoV-2), and 12 COVID negative samples, and identified all the clinical samples accurately.In terms of the scope, the current work demonstrates COVID19 mutation detection with an easy lateral flow-based readout, within ~75 minutes. While the assay has been shown to be able to detect 20 samples accurately, a larger data set might be needed to evaluate the sensitivity of the assay and the range of viral loads it can detect. The past year has seen a number of new techniques for COVID19 diagnosis, all of which will be useful for the still ongoing large-scale testing worldwide. In that context, this assay has practical and immediate use for COVID19 detection if it can be commercialized, especially with the rise of a number of variants of the SARS-CoV-2 virus.I liked reading the paper – it was very clear and crisp, and the data was well presented in both the text and the figures. However, as written, the paper seems to rely heavily on the reader's knowledge of their earlier work on FELUDA. From a reader's point of view, I felt that certain aspects of the assay, methodology and scope of immediate use could be expanded here even if it is a repeat from the authors' FELUDA work, to make this a stand-alone paper.

We thank Reviewer 3 for appreciating our work and commenting that the assay has practical and immediate use for COVID19 detection if it can be commercialized. The prime objective for this study is to develop an assay that can detect variants rapidly bypassing the cost and time associated with sequencing, and we are happy to report that we are in talks with interested parties for the global deployment of this assay.

We appreciate the Reviewer’s concerns about expanding more about the assay, methodology, and immediate use. We have now addressed each of these points in the revised version of the manuscript.

Discussion can be included on the following points. While these are not technical concerns, I think it will be informative for scientific and lay audience to know where the assay stands in terms of immediate impact for COVID19 pandemic.–Metrics of the assay such as sensitivity and specificity (even if referring to previous works).

We thank Reviewer 3 for this point. We have now performed this assay on a larger sample size and have included the sensitivity and specificity metrics of the assay in the revised version of the manuscript (Figure 4E, F).

– Scope of the assay: Do the authors intend to use this as a clinical test or at POC? Specific aspects can be discussed for either or both of these scenarios.

We intend to use this test for rapid early screening of variant signatures. The pipeline of the RAY assay requires a PCR machine and can therefore be done in clinical labs. It then again can be extended to become a POC test in combination with isothermal amplification. Since the FELUDA protocol has been shown to work with Recombinase Polymerase Amplification (RPA), we foresee that adapting RAY to POC-like settings is possible. As Reviewer 3 suggested, we have now discussed these scenarios in the manuscript.

– For the assay to be useful in POC setting, sample processing is an important step. Assuming samples are collected on swabs, RNA needs to be extracted before RAY can be performed. How can this be brought into the regular workflow for the assay to work at a testing site outside a lab or clinical facility? How will this affect the cost of the assay? The tally of the assay cost excludes RNA extraction. How can such an enzymatic approach be performed in a remote site without a cold chain?

The Reviewer has correctly raised the point about RNA extraction being required for the assay. Currently, the original version of the assay (called FELUDA) is being done in lab settings. Since RNA requires careful handling, we do not foresee how this can be avoided except where an automated system for RNA extraction and subsequent assay steps can be set up. If this is successful, the test can be performed outside a lab or clinical facility.

We have not included the cost of RNA extraction since the assay has been tested with samples obtained by both columns, automated as well as column-free extraction methods. Since the costs associated with each vary a lot, we have kept the extraction methods separate from the tally.